# A subcortical circuit linking the cerebellum to the basal ganglia engaged in vocal learning

Ludivine Pidoux*, Pascale Le Blanc, Carole Levenes, Arthur Leblois

Center for Neurophysics, Physiology and Pathology (UMR CNRS 8119), Centre National de la Recherche Scientifique (CNRS), Institute for Neuroscience and Cognition, Paris Descartes University, Paris, France

**Abstract** Speech is a complex sensorimotor skill, and vocal learning involves both the basal ganglia and the cerebellum. These subcortical structures interact indirectly through their respective loops with thalamo-cortical and brainstem networks, and directly via subcortical pathways, but the role of their interaction during sensorimotor learning remains undetermined. While songbirds and their song-dedicated basal ganglia-thalamo-cortical circuitry offer a unique opportunity to study subcortical circuits involved in vocal learning, the cerebellar contribution to avian song learning remains unknown. We demonstrate that the cerebellum provides a strong input to the song-related basal ganglia nucleus in zebra finches. Cerebellar signals are transmitted to the basal ganglia via a disynaptic connection through the thalamus and then conveyed to their cortical target and to the premotor nucleus controlling song production. Finally, cerebellar lesions impair juvenile song learning, opening new opportunities to investigate how subcortical interactions between the cerebellum and basal ganglia contribute to sensorimotor learning.
DOI: https://doi.org/10.7554/eLife.32167.001

*For correspondence:
ludivine.pidoux@gmail.com

**Competing interests:** The authors declare that no competing interests exist.

## Introduction

Speech is a highly complex motor skill which requires precise and fast coordination between vocal, facial and respiratory muscles. Human infants learn to reproduce adult vocalizations and to progressively master speech motor coordination within their first few years of life through an imitation process that builds up on motor sequence learning and strongly relies on auditory feedback (*Kuhl and Meltzoff, 1996*). This process, called vocal learning, is widely believed to rely on similar mechanisms as sensorimotor learning in general (*Doupe and Kuhl, 1999*; *Kuhl and Meltzoff, 1996*). The neural mechanisms underlying this process remain, however, poorly understood. Brain circuits known to be essential for sensorimotor adaptation and learning, namely the basal ganglia-thalamo-cortical loop (*Krakauer and Mazzoni, 2011*; *Pekny et al., 2015*) and the cerebello-thalamo-cortical loop (*Brooks et al., 2015*; *Izawa et al., 2012*), are both crucial for vocal learning in humans (*Vargha-Khadem et al., 2005*; *Ziegler and Ackermann, 2017*). The anatomical structure of these circuits and their function in sensorimotor learning are well conserved over vertebrate evolution (*Grillner and Robertson, 2016*; *Redgrave et al., 1999*; *Sultan and Glickstein, 2007*). In particular, avian song learning has been used as a paradigm to study the neural mechanisms of vocal learning, as it shares striking similarities with human speech learning (reviewed in *Doupe and Kuhl, 1999*).

The basal ganglia-thalamo-cortical network is involved in sensorimotor learning in several species, from lamprey to primates (*Hikosaka et al., 2002*; *Stephenson-Jones et al., 2013*; *Wickens et al., 2007*). The basal ganglia are thought to rely on reward prediction error signals conveyed by dopaminergic neurons (*Gadagkar et al., 2016*; *Schultz et al., 1997*; *Wickens et al., 2003*) to drive reinforcement learning strategies (*Doya, 2000*; *Sutton and Barto, 1981*). In songbirds, a specialized

**eLife digest** Human infants learn to speak by imitating the speech of adults around them. Over time, they learn to coordinate movements of their vocal cords and breathing muscles to produce specific sounds. Juvenile songbirds go through a similar process while learning to sing. Fledglings mimic adult birds and each other as they learn to produce their own songs. Songbirds are therefore often used as a model for how the brain drives vocal learning – whether of speech or song.

Circuits made up of similar brain regions support vocal learning in infants and in songbirds. These regions include areas of cortex, the outermost layer of the mammalian brain, as well as structures deep below the cortex. The latter include the basal ganglia, a set of structures that help mammals learn and perform fine motor skills.

But there is one brain region that has been implicated in vocal learning in infants but not in songbirds. Known as the cerebellum or 'little brain', this structure also helps with planning and performing movements. Anatomical studies in songbirds suggest a connection between the cerebellum and song-related circuits. But a direct role in birdsong has never been shown. Pidoux et al. now demonstrate that stimulating the cerebellum in anaesthetized zebra finches activates basal ganglia neurons involved in song learning. This activation spreads through a song-related circuit to neurons controlling the vocal cords. Disrupting the cerebellum, by contrast, makes it harder for juvenile birds to imitate adult song.

This is the first direct evidence for a role of the cerebellum in the acquisition of birdsong. Beyond vocal learning, the results shed light on the circuits that support motor learning more generally. They also suggest that we can use songbirds to study the cerebellum and its interactions with the basal ganglia. Abnormal interactions between these regions occur in movement disorders such as Parkinson's disease. Studying these interactions in the healthy mammalian brain should provide clues to the pathology behind these conditions.

DOI: https://doi.org/10.7554/eLife.32167.002

circuit homologous to the motor loop of the mammalian basal ganglia (*McCasland, 1987*; *Doupe et al., 2005*) is critical for song learning in juveniles and plasticity in adults (*Brainard and Doupe, 2002*). This circuit is thought to correct vocal errors through reinforcement learning driven by an internal song evaluation signal conveyed by dopaminergic neurons (*Fee and Goldberg, 2011*; *Gadagkar et al., 2016*; *Hoffmann et al., 2016*).

The cerebello-thalamo-cortical circuit also participates in sensorimotor learning in vertebrates, from fishes to primates (*Brooks et al., 2015*; *Gómez et al., 2010*; *Lewis and Maler, 2004*). It is believed to implement error-based supervised learning (*Albus, 1971*; *Ito, 1984*; *Knudsen, 1994*; *Marr, 1969*; *Raymond et al., 1996*) based on an error prediction denoting a mismatch between sensory prediction and actual sensory feedback (*Doya, 2000*; *Dreher and Grafman, 2002*). The cerebellum also drives on-line correction during movements building on the same sensory error prediction (*Tseng et al., 2007*) and controls the duration of movements and its prediction during sensorimotor learning (*Day et al., 1998*; *Flament and Hore, 1988*; *Izawa et al., 2012*). The existence of a pathway from the cerebellum to the song-related basal ganglia has been suggested by previous anatomical studies in songbirds (*Person et al., 2008*; *Vates et al., 1997*; *Nicholson et al., 2018*), but whether cerebellar circuits are involved in avian song learning and production remains unknown.

Beyond the indirect interaction via their respective loop with thalamo-cortical and brainstem networks, the basal ganglia and the cerebellum interact via a subcortical disynaptic pathway through the dentate nucleus, the motor part of the thalamus - more precisely the ventral anterior and ventral lateral nuclei of the thalamus in monkeys, and the centro-median nucleus of the thalamus in rodents - and the striatum (*Bostan et al., 2010*; *Chen et al., 2014*; *Hoshi et al., 2005*). The cerebellum and the basal ganglia therefore may not simply act in parallel to shape cortical and brainstem activity during learning. Instead, we hypothesize that cerebellar signals may reach the basal ganglia to drive error correction and reinforcement learning through the same output pathway. We test this hypothesis in zebra finches. We show that (i) cerebellar inputs are conveyed to the basal ganglia in songbirds

via the thalamus, (ii) they drive activity in the cortical target of the basal ganglia, and (iii) the cerebellar signals contribute to juvenile song learning, and to the timing of song elements.

## Results

To test the hypothesis that cerebellar signals are sent to the song-related basal ganglia circuits and that the cerebellum participates in song learning, we performed the following experiments. We first reproduced the anatomical finding by *Person et al. (2008)* showing that the deep cerebellar nuclei (DCN) send a projection to a thalamic region, which in turn projects to the song-related basal ganglia nucleus Area X. We then recorded responses to DCN electrical stimulation in Area X and its cortical targets and determined the nature of the neural pathway involved with pharmacological manipulations. Finally, we looked at the impact of lesions in the DCN on acoustic and temporal features such as syllable fundamental frequency, amplitude and duration, and compared song learning ability in juvenile finches following DCN or sham lesions.

### Anatomical connections exist from the DCN to the basal ganglia via the thalamus

We performed anatomical tracing experiments to confirm the previously reported (*Person et al., 2008*) indirect connection from the DCN to the song-related basal ganglia nucleus Area X, via the dorsal thalamic zone (DTZ). In a first set of experiments (n = 2 birds), we used two bidirectional tracers (fluorescently tagged dextran) injected both in Area X and the lateral DCN (*Figure 1A–C*). We then injected in Area X a retrograde tracer captured by synapses, Cholera-toxin B, while a bidirectional tracer (fluorescently tagged dextran) was injected in the lateral DCN (n = 1 bird, *Figure 1D–G*). In the cerebellum, the concomitant labeling of DCN and Purkinje cells indicated the proper location of the injection sites in the DCN (*Figure 1D* and *Figure 1—figure supplement 1*). As illustrated in both examples (*Figure 1B,C,E and F*), we found fibers labeled with the DCN-injected tracer in DTZ, posterior to the thalamic nucleus involved in song learning and production, the dorsolateral nucleus of the anterior thalamus (DLM). This provides evidence of axonal projections from the lateral DCN neurons to this region. Within the same DTZ area, cell somata of thalamic neurons were labeled with either the bidirectional or retrograde tracer injected in Area X (*Figure 1B,C,E and F*). We observed a close association between the two types of tracers with anterogradely-labeled fibers making putative contacts on retrogradely-labeled cell bodies (*Figure 1G*). This observation suggests that neurons in the lateral DCN project to DTZ thalamic neurons, which in turn project to Area X.

We also injected bidirectional tracers (fluorescently tagged dextran, n = 2 birds) in DTZ (*Figure 1H*). In the cerebellum, retrograde transport of the tracer was confined to large cell bodies within the DCN (*Figure 1I*). These large cells likely correspond to the large glutamatergic DCN output neurons that project to premotor areas. Labeled cell bodies were located for the most part in the lateral DCN. We did not find dorso-ventral distinction in the labelling of the lateral DCN, suggesting that the projection from the lateral DCN to DTZ is not topographically organized (*Figure 1I*). Some neurons in the *interpositus* nucleus were also labeled (results not shown). This suggests that, even if the projection from the cerebellum to DTZ largely comes from the lateral DCN, the *interpositus* may also be partially involved in this cerebello-thalamic projection. Regarding the anterograde transport of tracers injected in DTZ (*Figure 1H*), we found many labeled axonal fibers in Area X, confirming the direct projection from DTZ to Area X (*Figure 1J*).

Thus, as already suggested in a previous study (*Person et al., 2008*), we found anatomical evidence for a disynaptic connection between the cerebellum and the song-related basal ganglia Area X: the lateral DCN sends projections to DTZ which in turn projects to Area X. Importantly, these anatomical results have been replicated very recently, confirming the existence of the DCN-DTZ-Area X pathway (*Nicholson et al., 2018*).

### The connection from DCN to basal ganglia is functional

We then determined whether this DCN-DTZ-Area X pathway drives activity within the basal ganglia. To this end, we investigated the responses evoked by DCN electrical stimulation in Area X neurons in anaesthetized zebra finches. To this end, we investigated the responses evoked by DCN electrical stimulation in Area X neurons.

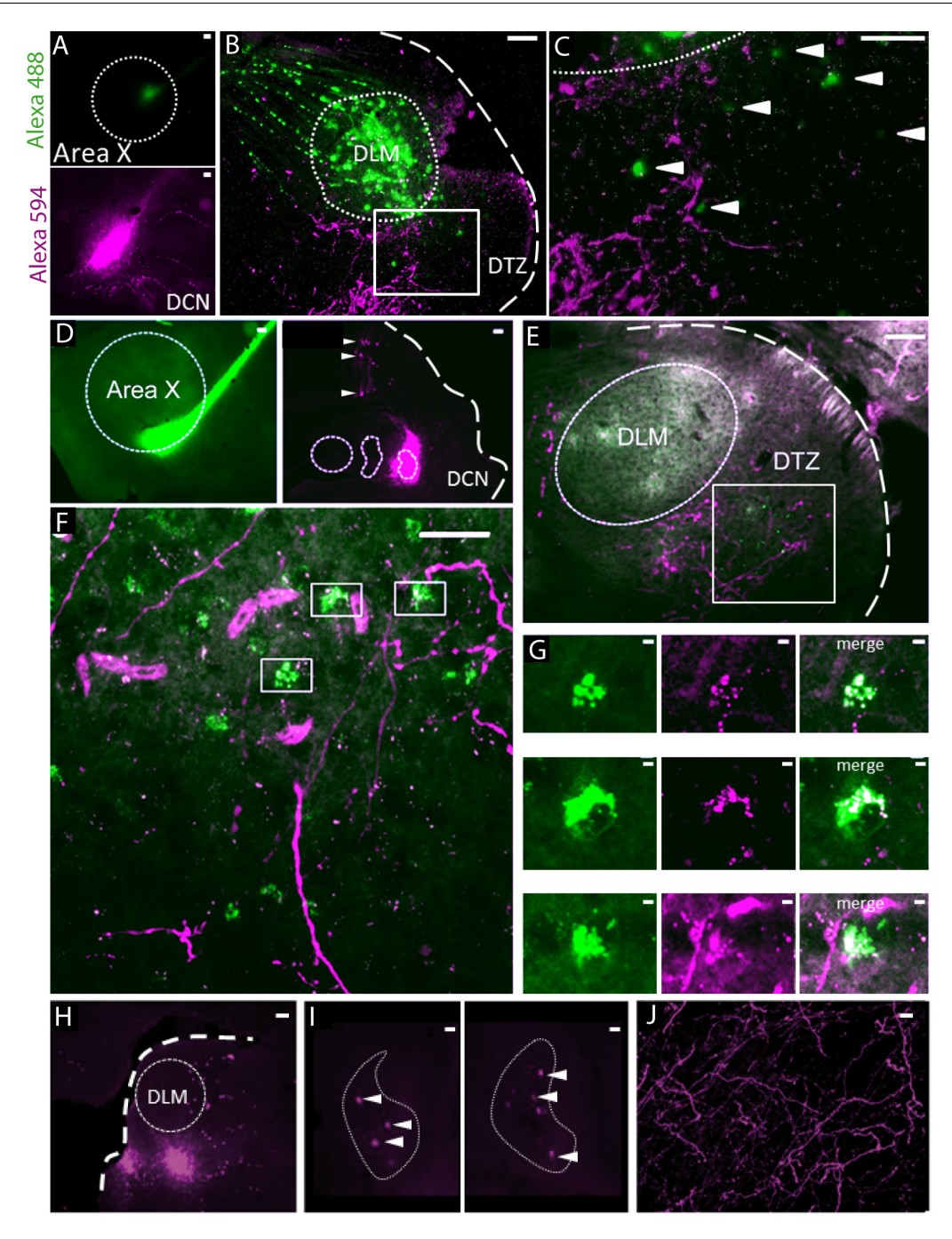

**Figure 1.** Anatomical connection between DCN and Area X. (**A**) Injection sites of Dextran Alexa 488 (green, top panel) and Dextran Alexa 594 (magenta, bottom panel, sagittal sections). Dotted line delimits Area X (top panel). Scale bar: 100 μm. (**B**) Labelling in the dorsal thalamic zone (DTZ) showing efferent cerebellar fibers (magenta) and cell bodies of neurons in DTZ (green). Large labelling of efferent fibers from Area X nucleus was also found in DLM, as the tracer is anterogradely and retrogradely transported. The dotted line in B delimits nucleus DLM, while the white square indicates magnification location for C. The large dotted line delimits the brain slice contour. Scale bar: 100 μm. DLM: dorsolateral nucleus of the anterior thalamus. (**C**) Magnification of the dorsal thalamic zone. Cerebellar fibers are labelled in magenta, and somata are labelled in green and indicated with arrowheads. Scale bar: 100 μm. (**D**) Injection sites of cholera toxin B in Area X (green, left panel) and Dextran Alexa 594 in DCN (magenta, right panel). Dotted lines delimit Area X (left panel) and all three DCN (right panel). The large dotted line delimits the brain slice contour. Retrograde labeling of Purkinje cells projecting to the DCN targeted by dye injection can be observed (right panel, arrowheads). Scale bar: 100 μm. (**E–F**): Close contacts were observed in the dorsal thalamic zone (DTZ, scale bars: 100 μm). The dotted line in E delimits nucleus DLM, while the white square in E and F indicates magnification location. Efferent fibers from Area X in DLM appear as diffuse green labeling in this nucleus, while green cell somas in DTZ reflect

*Figure 1 continued on next page*

*Figure 1 continued*

afferent neurons. Magenta-labeled fibers from the DCN surround Area X-projecting neurons in DTZ. (**G**): Three examples of close contacts between fibers from the DCN (magenta, middle panel) and soma of neurons projecting to Area X (green, left panel) in DTZ. Each panel in G corresponds to a magnification of squares indicated in F. The merge suggests an anatomical connection (right panel). Scale bar: 2 µm. (**G**) Injection sites of Dextran Alexa 594 in DTZ. The large dotted line delimits slice contours, and the dotted circle represents DLM. Scale bar: 100 µm. (**H**) Two examples of retrograde labelling in the lateral DCN following DTZ injection showed in G. Both examples are from the same animal, at two different depths. (**I**) Arrowheads indicate DCN cell soma labelled. The dotted line delimits the lateral DCN contours. Scale bar: 20 µm (**J**) Example of anterograde labelling in Area X. Only fibers (but no soma) were observed in Area X after DTZ injection. Scale bar: 2 µm.

DOI: https://doi.org/10.7554/eLife.32167.003

The following figure supplement is available for figure 1:

**Figure supplement 1.** Magnification of Purkinje cell labelling.

DOI: https://doi.org/10.7554/eLife.32167.004

Most neurons are silent or display very little spontaneous activity in Area X under isoflurane anesthesia, whereas a minority of them displays high spontaneous activity (>25 spikes/sec, see Materials and methods). These spontaneously active neurons are most likely pallidal-like neurons (*Leblois et al., 2009*; *Person and Perkel, 2007*). Hereafter, this population of neurons, at least some of which are Area X projection neurons (*Goldberg et al., 2012*; *Leblois et al., 2009*), will be referred as pallidal neurons. DCN stimulation provoked a strong increase in the firing rate of pallidal neurons, as illustrated in *Figure 2B*. When a response was evoked by single-pulse stimulation in at least one pallidal neuron in Area X, all subsequently recorded neurons were also responsive to the stimulation. However, their response profile at a given intensity differed from one another. This diversity of response profiles could be classified as follows: single excitatory responses (observed in 71% of case, *Figure 2C*, two last bottom panels 0.2 and 0.5 mA), biphasic responses with excitation followed by inhibition (observed in 19% of case, *Figure 2C*, middle panel 1 mA), or triphasic responses with a rapid inhibition followed by an excitation and a late inhibition (observed in 10% of case, *Figure 2C*, top panel 2 mA). Interestingly, different response profiles were found in the same neuron depending on the stimulation intensity used: higher stimulation intensity induced biphasic or triphasic responses, while lower stimulation intensity only caused excitation. Previous studies have shown that excitatory inputs to Area X can drive such biphasic or triphasic responses in pallidal neurons due to feedforward inhibition mediated by local Area X inhibitory neurons (*Leblois et al., 2009*). The response latencies between the onset of the stimulation pulse and the onset of the excitatory response (see Materials and methods) were broadly distributed from 10 to 50 ms ($20.80 \pm 4.56$ ms, median: 21 ms, *Figure 2D*). While short latency responses (10–20 ms) can be naturally explained by a disynaptic excitatory transmission from the DCN to Area X through DTZ, biphasic and triphasic responses involve longer latencies to which feedforward inhibition within Area X likely participates. Indeed, fast feedforward inhibition within Area X can delay the response of pallidal neurons to their excitatory inputs (*Leblois et al., 2009*), as it is the case in the mammalian striatum (*Mallet et al., 2005*). Altogether, these results show that stimulation of DCN neurons can drive the activity of pallidal neurons in Area X, confirming that the latter receive a functional input from the cerebellum.

## Note added in proof

In the original version of the paper, the stimulation intensities applied in the deep cerebellar nuclei (DCN) to evoke neural responses in Area X were high relative to the intensities typically used in ortho- or anti-dromic functional mapping. The duration of stimulation pulses was also long (1 ms), leading to a high level of total stimulation current. The selectivity of such stimulation may therefore be questioned. To resolve this issue, we have pursued additional experiment to assess the responsiveness of Area X pallidal neurons following low intensity (50-200 µA) and short duration (100 µs) stimulation pulses in the lateral DCN. We show in a new figure (*Figure 3* of the present version of the article) that Area X pallidal neurons show strong responses to these low-intensity and short-duration stimulation. Moreover, we provide evidence that even high-intensity (1 mA) and long-duration (1 ms) stimulation, when applied a few hundreds of microns away from the lateral DCN, do not evoke responses in Area X pallidal neurons. Altogether, our additional results confirm that pallidal neurons in Area X selectively respond to the stimulation of the lateral DCN.

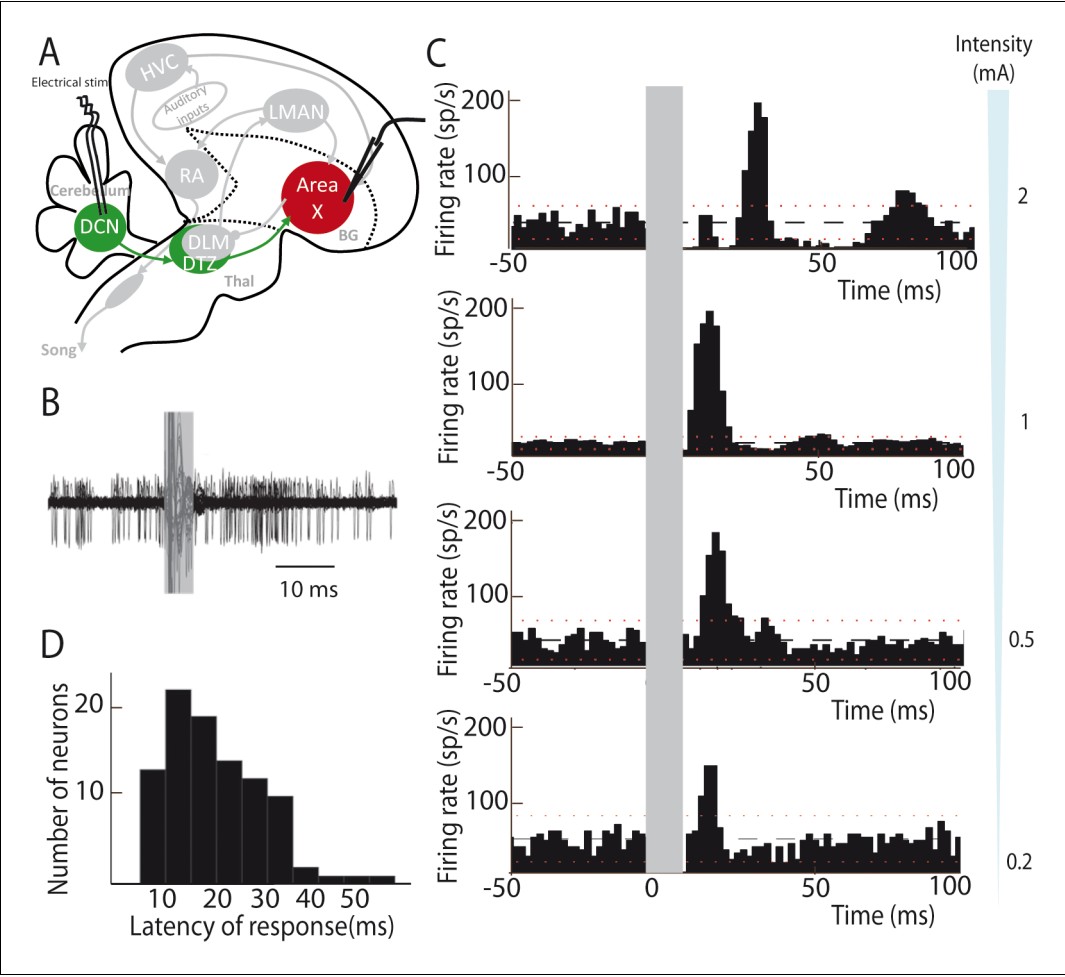

**Figure 2.** Deep cerebellar stimulation elicits strong excitation in pallidal cells of Area X. (**A**) Diagram of the song system in songbirds. In all song system diagrams in *Figures 2–6*, we highlight nuclei involved in the experiment reported on a given figure in color, while other song system nuclei are in grey shades. The colors used are different for each pathway: red for the song-related **basal ganglia**-thalamo-cortical circuit composed of the basal ganglia nucleus Area X, the thalamic nucleus DLM, and the cortical nucleus LMAN, green for the cerebello-thalamo-**basal ganglia** circuit through the DCN and DTZ, and black for the motor pathway composed of HVC and RA. Here, stimulations were performed in the DCN during the recording of pallidal neurons in Area X. HVC: used as a proper name, RA: robust nucleus of the archopallium, LMAN: lateral magnocellular nucleus of the anterior nidopallium, DLM: medial portion of the dorsolateral nucleus of the anterior thalamus, DTZ: dorsal thalamic zone, DCN: deep cerebellar nuclei. (**B**) Twenty superimposed extracellular recording traces around DCN stimulation show the increase in the number of spikes produced by a representative pallidal neuron following DCN stimulation (grey rectangle) compared to baseline firing. (**C**) Peri-stimulus-time-histograms (PSTHs) representing the firing rate of 4 different pallidal neurons around DCN stimulation (time bin: 2 ms). The black horizontal dashed line depicts the mean baseline firing rate and red dotted lines indicate confidence intervals (2.5 SD away from the mean baseline firing rate). Different response profiles are shown: excitation only (the two in the bottom, stimulation at 0.2 and 0.5 mA), biphasic response (second PSTH from top, stimulation at 1 mA), or inhibition and biphasic response (top, stimulation at 2 mA). (**D**) Distribution of response latencies between DCN stimulation and the beginning of the excitatory response (20.80 ± 4.56 ms, median: 21 ms).

DOI: https://doi.org/10.7554/eLife.32167.005

The following source data is available for figure 2:

**Source data 1.** Peri-stimulus time histogram (PSTH) code.

DOI: https://doi.org/10.7554/eLife.32167.006

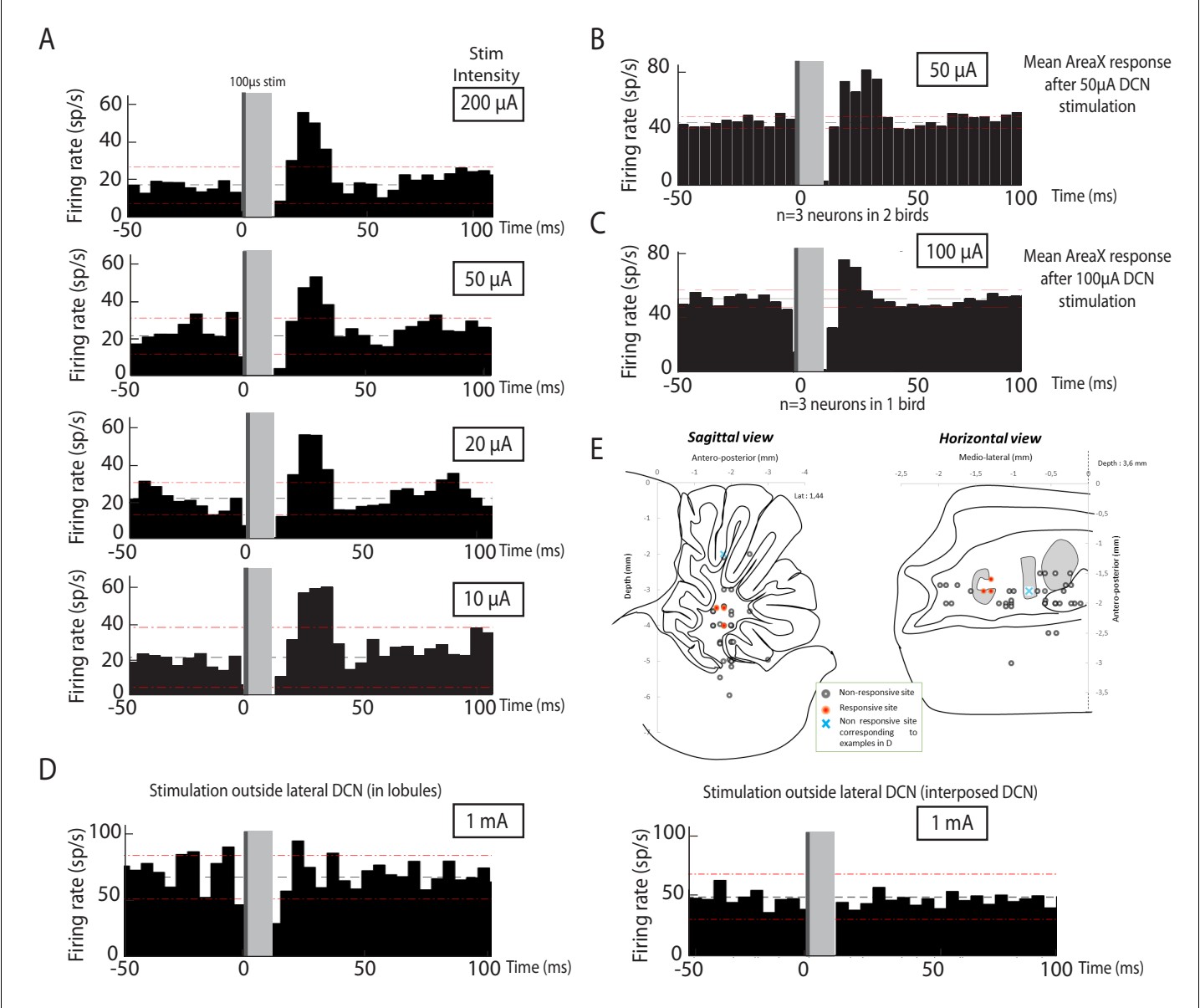

**Figure 3.** Area X neurons selectively respond to electrical stimulation of the lateral deep cerebellar nucleus (DCN), even at low intensity. Following low-intensity (from 50 to 200µA) stimulation of the lateral DCN through short-duration (100 µs) electrical pulses induces responses in pallidal-like neurons of the song-related basal ganglia nucleus Area X. (A) Response to DCN stimulation at various current intensities ranging from 50 to 200 µA in a typical pallidal-like neuron from Area X (100 µs monophasic pulses, represented by dark grey rectangles, while light grey rectangles indicate the 'blind' period due to stimulation artifact). With each stimulation intensity, we observe an excitation of pallidal-like neurons following stimulation in the lateral DCN. The black dotted line represents the mean frequency of the pallidal-like neuron during the baseline period, and the red dotted line corresponds to 2.5*standard deviation from baseline. Excitation/inhibition is significant when two consecutive columns are upper/lower than red lines. Time bin: 5 ms. (B) 50µA DCN stimulation induces an increased firing rate in pallidal-like neurons. In n=3 neurons recorded in two different birds, we observed a similar increase in activity following 50 µA stimulation (100 µs monophasic pulses, time bin: 5 ms). (C) 100µA DCN stimulation induces an increased firing rate in pallidal-like neurons. In n=3 neurons recorded in two different birds, we observed a similar increase in activity following 100 µA stimulation. (100 µs monophasic pulses, time bin: 5 ms). (D) Area X pallidal-like neurons do not respond to electrical stimulation outside the lateral DCN. The pallidal-like neurons did not exhibit significant inhibition or excitation when the electrical stimulation was applied away from the lateral DCN even at high intensity (1 mA, 1 ms-long monophasic pulses), neither in the cortex (left panel, coordinates indicated in E by a cross) nor in another DCN (stimulation in interposed nucleus, right panel, coordinates indicated in E by a cross). (E) Responses in Area X pallidal-like neurons were limited to stimulation points located in the lateral DCN. Coordinates used to place the stimulation electrode were summed up on two different schemes corresponding to sagittal (left) or horizontal (right) views. Only a few points (n=3) of stimulation induced responses in pallidal-like neurons (red point). Stimulations at other coordinates (black point, n =32 points) did not induce any response. Blue crosses correspond to the two examples showed in D.

DOI: https://doi.org/10.7554/eLife.32167.007

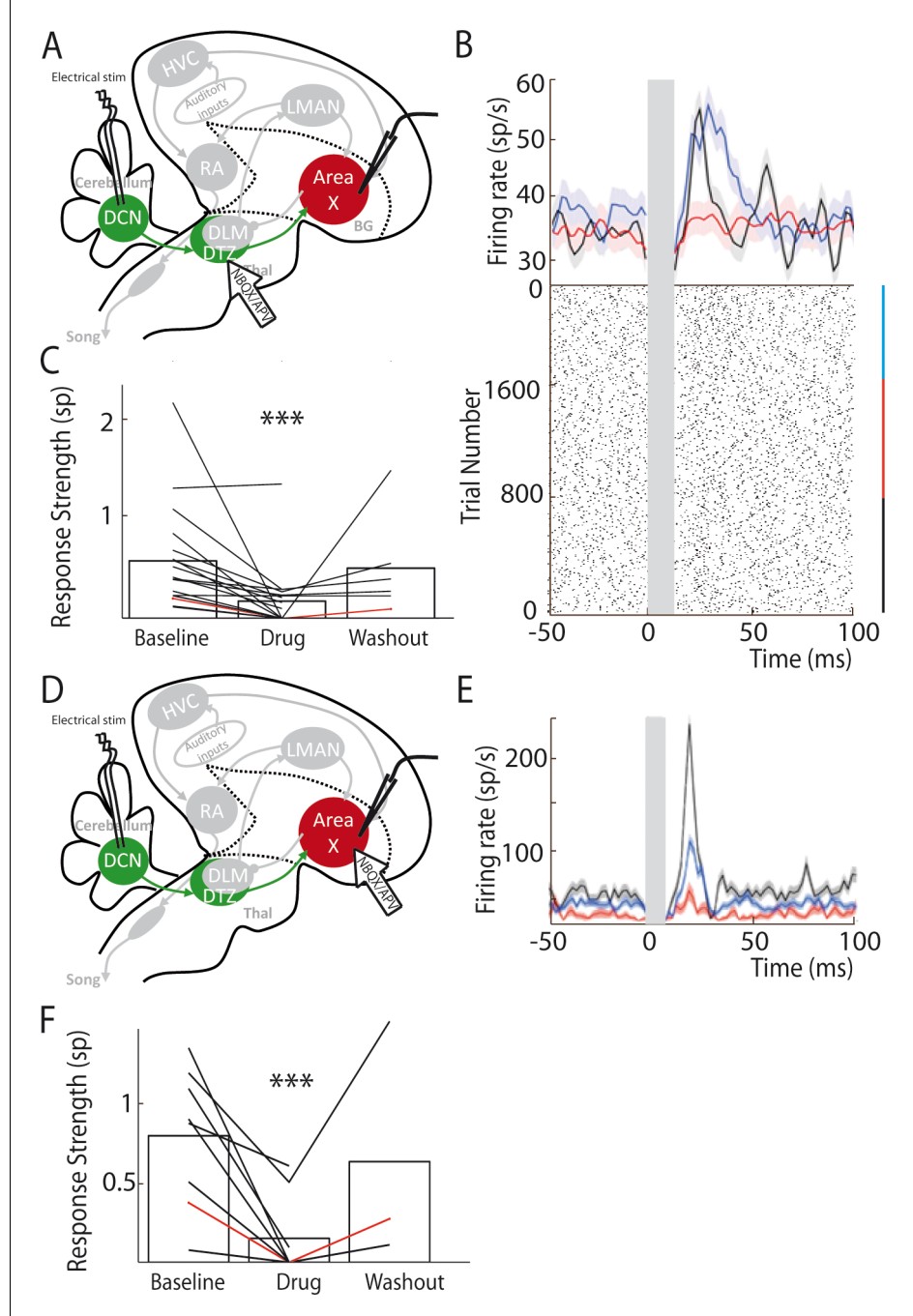

**Figure 4.** Area X pallidal responses to DCN stimulation are transmitted through excitatory synapses in DTZ and Area X. (A) Diagram of the song system in songbirds, as in *Figure 2A*. Recordings were performed in Area X, NBQX/APV were applied in DTZ. (B) PSTH (top part) of a typical pallidal neuron before (black), during (red) and after (blue, washout) drug application in DTZ, and their corresponding raster plots (bottom part). (C) Population data showing the response strength of pallidal neurons in the three conditions (baseline, drug and washout, n = 16 pallidal neurons in 8 birds, paired Wilcoxon test, p value<0.001). The red line represents the example shown in B. (D) Diagram of the song system, as in *Figure 2A*. Recordings were performed in Area X, NBQX/APV were applied in Area X in proximity to the recorded neuron. (E) PSTH representing the firing rate of one pallidal neuron, before (black), during (red) and after (blue, washout) drug application in Area X. Baseline activity after drug application (red) sometimes slightly decreases in Area X neurons compared to before drug application (black), but no significant change was observed over all neurons recorded in this condition (see Results). (F) Population data showing the evolution of response strength before, during and after drug application (n = 8

*Figure 4 continued on next page*

*Figure 4 continued*
pallidal neurons in 7 birds, paired Wilcoxon test, p value<0.001). The red curve represents the example shown in E. In this figure and the following ones, stars indicate significance level (*p<0.05; **p<0.01; ***p<0.001).
DOI: https://doi.org/10.7554/eLife.32167.008

## The thalamic region DTZ mediates the cerebello-basal ganglia pathway

Our anatomical results suggest that DTZ relays Area X neuronal responses to cerebellar stimulation. To demonstrate this, we blocked glutamatergic transmission in DTZ while monitoring the responses in Area X to DCN stimulation. In mice and rats, the cerebellar projections to the thalamus are mediated by glutamate (*Kuramoto et al., 2009*; *Kuramoto et al., 2011*). We therefore pressure-injected a cocktail of AMPA/kainate and NMDA receptor antagonists, respectively 2,3-dihydroxy-6-nitro-7-sulfamoyl-benzo quinoxaline-2,3-dione (NBQX) and 2-amino-5-phosphonovaleric acid (APV), to block glutamatergic transmission within DTZ (see Materials and methods, *Figure 4A*). *Figure 4B* shows an example of the change in the response of a pallidal neuron to DCN stimulation following the injection of glutamatergic blockers in DTZ. As our hypothesis predicts, the excitation that DCN stimulation induced in this pallidal neuron was suppressed following drug injection. We then quantified the change in response induced by glutamatergic blockers in DTZ over the population of pallidal

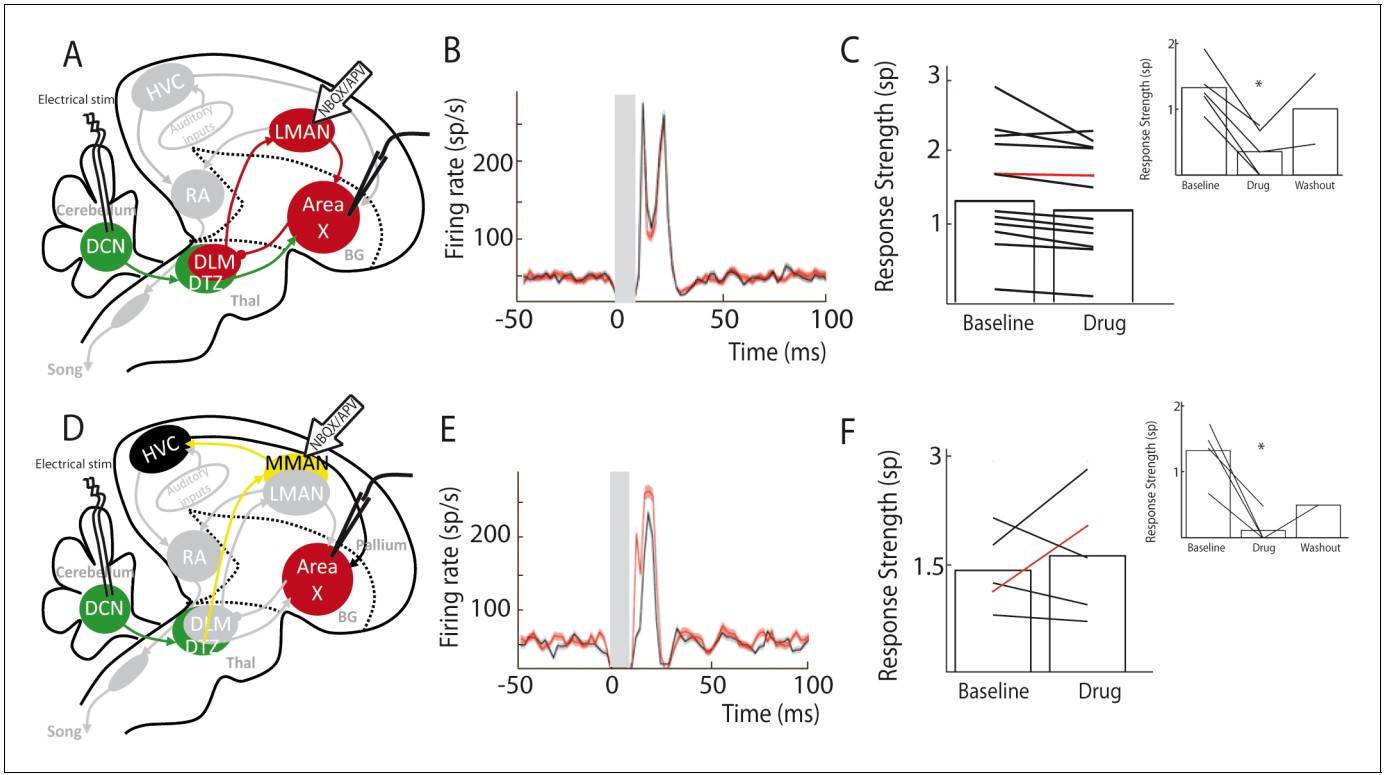

**Figure 5.** Area X pallidal responses to DCN stimulation are not transmitted through the cortical nuclei LMAN or MMAN. (**A**) Diagram of the song system, as in *Figure 2A*. Recordings were performed in Area X, NBQX/APV were applied in LMAN. (**B**) PSTH representing the firing rate of a pallidal neuron around DCN stimulation before (black) and during (red) drug application in LMAN. (**C**) Population data showing no change in response strength before and during LMAN glutamatergic blockade (n = 12 pallidal neurons in 6 birds, paired Wilcoxon test, non-significant). The red curve represents the example shown in B. Inset: confirmation of drugs efficiency by applying drugs around the recorded pallidal neuron (n = 5 pallidal neurons in 5 birds, paired Wilcoxon test, p<0.01). (**D**) Diagram of the song system. Recordings were performed in Area X, NBQX/APV were applied in MMAN, a nucleus projecting to HVC. (**E**) PSTH representing the firing rate of pallidal neuron before (black) and during (red) drug application in MMAN. (**F**) Population data showing the evolution of response strength before and during glutamatergic blockade in MMAN (n = 5 pallidal neurons in 2 birds, paired Wilcoxon test, non-significant). The red curve represents the example shown in E. Inset: confirmation of drug efficiency by applying drug on the recorded pallidal neuron (n = 5 pallidal neurons in 2 birds, paired Wilcoxon test, p<0.05).
DOI: https://doi.org/10.7554/eLife.32167.009

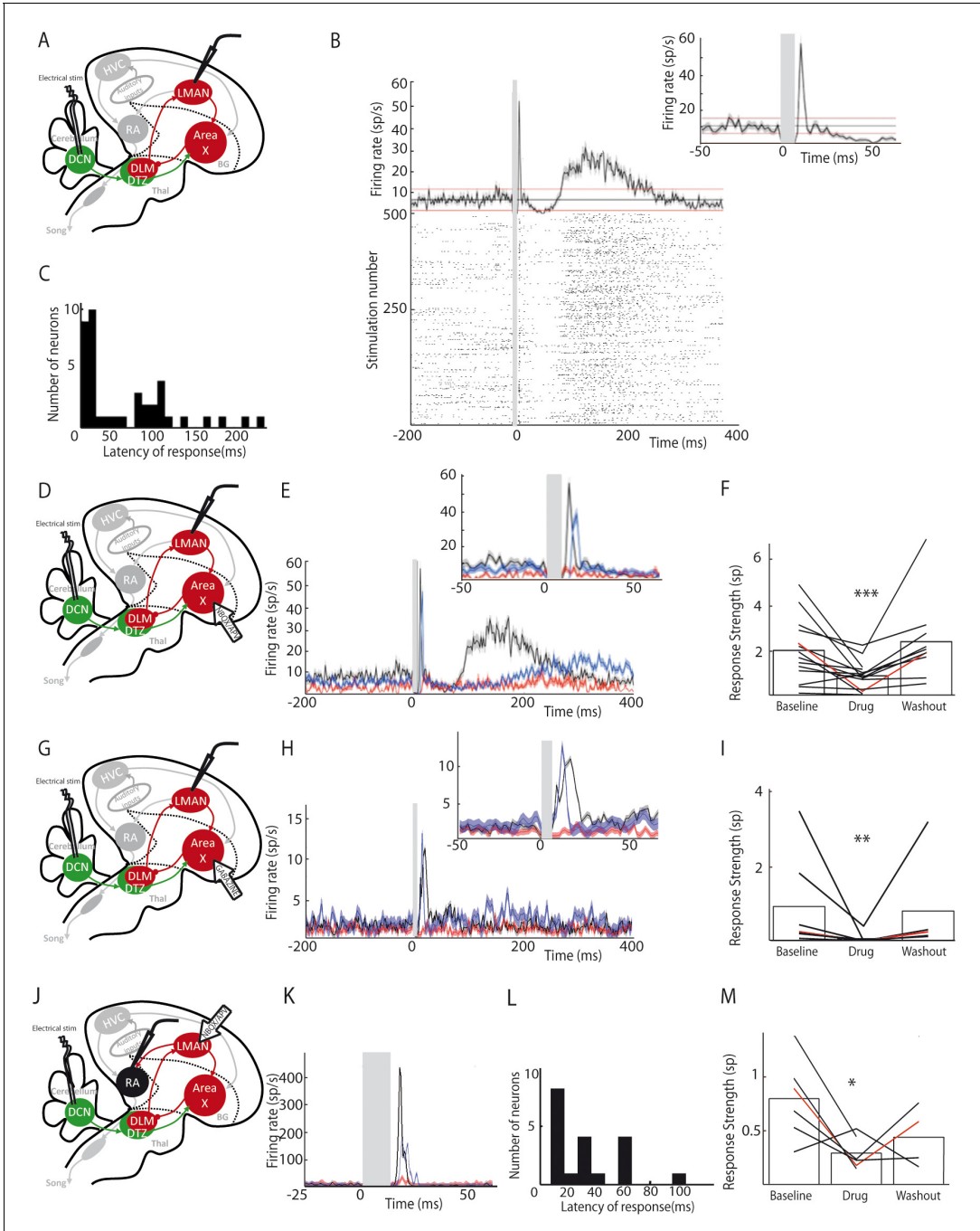

**Figure 6.** LMAN and RA neurons display responses to DCN stimulation. (**A**) Diagram of the song system, as in *Figure 2A*. Neuronal activity was recorded in LMAN during DCN stimulation (**B**) Example response in a LMAN multi-unit recording displaying two excitatory peaks following DCN stimulation with the corresponding raster plot. Inset: magnification of the first excitatory peak. (**C**) Distribution of response latency over all LMAN recordings displaying the two characteristic peaks of response (first peak: 10–30 ms and second peak: 100 ms, see Results, time bin: 10 ms). (**D**) Diagram of the song system. NBQX/APV were applied in Area X and neuronal activity was recorded in LMAN during DCN stimulation. (**E**) Example response following DCN stimulation from a multi-unit recording in LMAN, before (black), during (red) and after (blue, washout) the drug application. (**F**) Population data showing the evolution of response strength over the three periods (baseline, drug, washout, n = 14 multiunit recording sites in 5 birds, paired-Wilcoxon test, p value=0.001). The red curve represents the example shown in E. (**G**) Diagram of the song system. Gabazine was applied in Area X and neurons were recorded in LMAN during DCN stimulation. (**H**) Example response from a multi-unit recording in LMAN following DCN stimulation, before (black), during (red) and after (blue, washout) gabazine application. (**I**) Population data showing the evolution of the response strength over the three periods (baseline, drug, washout, n = 7 multiunit recording sites in 4 birds, paired-Wilcoxon test, p value=0.0156). The red curve represents the example shown in H. (**J**) Diagram of the song system, as in *Figure 2A*. Neurons were recorded in RA during DCN stimulation, NBQX/APV were applied

*Figure 6 continued on next page*

*Figure 6 continued*
in LMAN. (**K**) PSTH representing the firing rate of a typical RA neuron before (black), during (red) and after (washout, blue). (**L**) Distribution of RA neurons response latencies (time bin: 10 ms). (**M**) Population data showing the change of response strength over the three periods (baseline, drug, washout, n = 6 neurons in 5 birds, paired Wilcoxon test, p<0.05). The red curve represents the example shown in C.
DOI: https://doi.org/10.7554/eLife.32167.010

neurons recorded under this pharmacological protocol (n = 16 pallidal neurons in 8 birds). The response strength and peak of the excitatory response (see Methods) were strongly reduced or totally suppressed when we blocked DTZ glutamatergic relay. Mean response strength decreased from 0.55 ± 0.13 spikes at baseline to 0.16 ± 0.04 spikes following drug injection (paired Wilcoxon test, p=3e-004, *Figure 4C*), and mean excitation peak decreased from 99 ± 23 Hz at baseline to 44 ± 10 Hz following drug injection (paired Wilcoxon test, p=5e-004). These results confirm that the responses to DCN stimulation in Area X pallidal neurons are relayed by glutamatergic transmission in DTZ.

Thalamo-striatal projections are glutamatergic in most vertebrates (*Smith et al., 2004*). It is thus natural to suppose that DTZ neuronal projections excite Area X neurons through glutamatergic transmission in zebra finches. We tested this hypothesis by blocking glutamatergic transmission around the recorded pallidal neuron upon injection of the same glutamatergic blockers as above (*Figure 4D*). We indeed confirmed that responses of pallidal neurons to DCN stimulation were abolished by the drug injection (*Figure 4E and F*; response strength decreased from 0.8 ± 0.3 spikes at baseline to 0.16 ± 0.05 spikes following drug injection, paired Wilcoxon test, p=0.008, and mean excitation peak decreased from 125 ± 44 Hz at baseline to 30 ± 10 Hz following drug injection, paired Wilcoxon test, p=0.008).

## LMAN does not mediate Area X responses to DCN stimulation

We cannot completely exclude that drugs injected in DTZ could diffuse to DLM, which would block a response mediated by the well-known DLM-LMAN-Area X pathway. To rule this alternative hypothesis out, we applied in LMAN the cocktail of AMPA and NMDA receptor antagonists while monitoring pallidal responses to DCN stimulation (*Figure 5A*). We found no significant difference in the excitatory response of pallidal neurons to DCN stimulation between baseline and drug application conditions (*Figure 5B and C*, n = 12 neurons in 6 birds; response strength was 1.49 ± 0.5 spikes at baseline and 1.34 ± 0.38 spikes following drug injection, paired Wilcoxon test, p=0.5; mean excitation peak was 211 ± 50 Hz at baseline and 200 ± 47 Hz following drug injection, paired Wilcoxon test, p=0.5). While LMAN does not appear to mediate the main response to DCN stimulation in Area X pallidal neurons, it may participate to a reverberation of the responses through the Area X – DLM – LMAN loop. In this respect, is interesting to note that all but one pallidal neurons underwent a slight decrease in their response upon glutamatergic blockade in LMAN, possibly reflecting a reduced reverberation in the loop. As the measured response strength only reflects the first peak of excitatory response in Area X, the slow response mediated by the propagation through the loop is unlikely to provide an important contribution to this measure (see Methods). Following each drug injection in LMAN, we verified the efficacy of the pressure injection by moving the drug pipette in the vicinity of the recorded pallidal neuron (*Figure 5C*, inset, n = 5 pallidal neurons in 5 birds). During those controls, DCN stimulation response strength decreased from 1.32 ± 0.59 spikes at baseline to 0.35 ± 0.16 spikes following drug injection (paired Wilcoxon test, p=0.008) and mean excitation peak was reduced from 182 ± 82 Hz at baseline to 57 ± 26 Hz following drug injection (paired Wilcoxon test, p=0.02). This confirms that our pharmacological blockades were efficient, and we can therefore rule out a transmission from DCN to Area X through the DLM-LMAN pathway.

## MMAN is not involved in Area X responses to DCN stimulation

In songbirds, the DTZ relays projections from the DCN to Area X and is composed of several thalamic regions as described previously by anatomical studies (*Person et al., 2008*; *Vates et al., 1997*). One of these regions, called the dorsal medial posterior thalamic zone (DMP) projects to the medial part of the magnocellular nucleus (MMAN) (*Foster et al., 1997*; *Nicholson et al., 2018*). MMAN is in turn implicated in a pathway ending in the song-related motor nuclei HVC (used as a

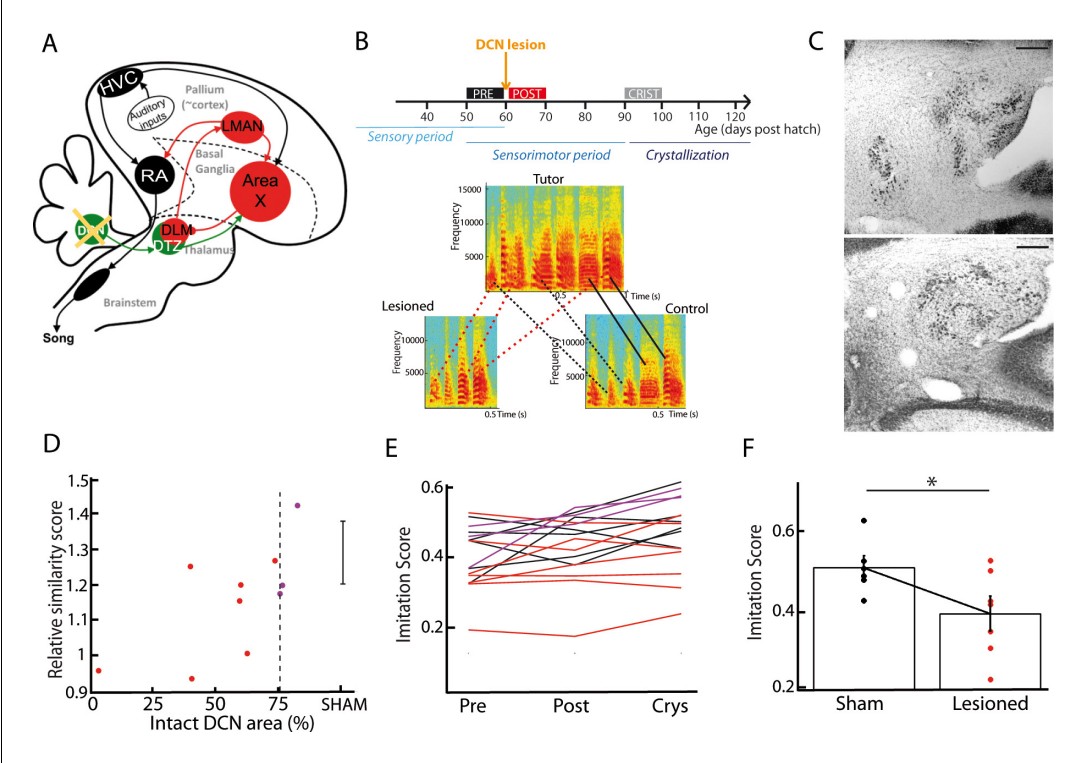

**Figure 7.** DCN lesions impair song learning and reduce the similarity to tutor song after crystallization. (**A**) Diagram of the song system, as in *Figure 2A*, representing DCN lesion. (**B**) Top: Diagram of the song learning periods in songbirds: the sensory period, the sensorimotor period in which juveniles start to produce sounds, and the crystallization phase. Lesions were made at 60 dph. Bottom: Examples of three spectrograms of tutor and pupil song motifs at crystallization: top: tutor, bottom left: a pupil with DCN lesion, bottom right: control pupil. Solid lines connect two similar syllables found in the tutor and juvenile song motifs, dotted lines between two syllables reflect a partial copy of the tutor syllable (red lines for the juvenile with DCN lesion, black lines for the control juvenile). (**C**) Nissl staining on horizontal slices showing the deep cerebellar nuclei. Top: control bird. Bottom: bird with DCN lesion (Scale bar: 100 μm). (**D**) Normalized imitations score (imitation score at crystallization divided by the imitation score before surgery) plotted as a function of the total area left from the lateral DCN (%) for juveniles with DCN lesion (red and purple dots, n = 10 birds). The mean and SD of the normalized imitation score over the sham group are represented as an error bar (n = 6 sham birds). For the following analysis, we consider the birds with a significant lesion (<75% of intact lateral DCN, vertical dotted line) represented with red dots. A significant correlation was revealed between the similarity and the proportion of lateral DCN left intact (r = 0.68, p<0.05). (**E**) Population data showing the evolution of the imitation score between the day before the lesion (pre) and the crystallization period (90 dph) in juveniles with sham lesions (black lines for individual birds) and DCN lesions with larger lesion size (red lines) or small lesion size (purple lines). (**F**) The imitation score at crystallization is significantly larger in the sham group than in the DCN large lesion group (n = 7 birds with large lesion, n = 6 sham birds, Wilcoxon test, p<0.05).
DOI: https://doi.org/10.7554/eLife.32167.011

The following source data and figure supplements are available for figure 7:

**Figure supplement 1.** Song rate is not affected by DCN lesion.
DOI: https://doi.org/10.7554/eLife.32167.012

**Figure supplement 2.** Example spectrogram of tutor and pupil (control and lesion) song motifs as a function of lesion size.
DOI: https://doi.org/10.7554/eLife.32167.013

**Figure supplement 3.** Effect of DCN lesions revealed by a custom-written similarity score based on the peak cross-correlation between the spectrograms of the tutor's motifs and of the pupil's songs.
DOI: https://doi.org/10.7554/eLife.32167.014

**Figure supplement 3—source data 1.** Source code for similarity score analysis.
DOI: https://doi.org/10.7554/eLife.32167.015

proper name) and RA (*Williams et al., 2012*). As HVC projects to Area X (*Nottebohm et al., 1976*; *Nottebohm et al., 1982*), we wondered whether the response we observed in Area X could be conveyed through this MMAN-HVC-X pathway. To rule out this possibility, we blocked glutamatergic transmission in MMAN while monitoring pallidal responses to DCN stimulation (*Figure 5D*). We found no significant effect of the drug injection in MMAN on the responses of pallidal neurons to

DCN stimulation (*Figure 5E and F*, n = 5 pallidal neurons in 2 birds, response strength was 1.43 ± 0.24 spikes at baseline and 1.63 ± 0.43 spikes following drug injection, Wilcoxon test, p=0.8; mean excitation peak was 246 ± 110 Hz at baseline and 259 ± 116 Hz following drug injection, paired Wilcoxon test, p=0.4). As previously, we checked the efficacy of the pressure injection through the glass pipette in Area X at the end of each experiment (*Figure 5F*, inset, n = 5 pallidal neurons, decrease from 1.33 ± 0.66 spikes at baseline to 0.12 ± 0.06 spikes following drug injection, Wilcoxon test, p=0.03; mean excitation peak decreased from 239 ± 119 Hz at baseline to 35 ± 18 Hz following drug injection, paired Wilcoxon test, p=0.03). This experiment ruled out the possible transmission from DCN to Area X pallidal neurons through the MMAN-HVC-Area X pathway.

In conclusion, the results of our electrophysiological experiments provide strong evidence that the cerebellum is linked to the song-related basal ganglia nucleus Area X through a functional excitatory connection involving a glutamatergic projection from the DCN to DTZ, and a glutamatergic projection from DTZ to Area X.

## The cerebellar responses are conveyed to LMAN and RA through the basal ganglia loop

In songbirds, Area X is known to be part of the basal ganglia-thalamo-cortical circuit homologous to the motor loop of the basal ganglia-thalamo-cortical networks in mammals (*Brainard and Doupe, 2002*). This basal ganglia-thalamo-cortical loop affects song production and drives song learning and plasticity via its projection to the premotor nucleus RA (*Andalman and Fee, 2009*; *Bottjer et al., 1984*). In the following experiments, we tested whether the responses observed in the pallidal neurons after DCN stimulation were conveyed to the output nucleus of the basal ganglia-thalamo-cortical loop, LMAN, and its efferent premotor nucleus RA (*Figure 6*).

DCN stimulation elicited strong responses in LMAN neurons (*Figure 6B*). Those responses could be composed of two excitatory components: a first strong and rapid one followed by a delayed and slow one. Such bimodal responses with two peaks were found in 10% (n = 3/30) of the LMAN neurons recorded. The remaining LMAN neurons (90%, n = 27/30) displayed one of the two excitatory responses provoked by DCN stimulation. The latency of excitatory responses in LMAN neurons was therefore spread in a bimodal distribution (*Figure 6C*) with two distinct peaks: a first one between 10 and 50 ms (26 ± 7.8 ms, median: 19 ms, 28% of all recorded LMAN neurons, n = 8/30), and a second one around 100 ms (125 ± 32 ms, median: 110 ms, 72% of all recorded LMAN neurons, n = 22/30). Interestingly, these two peaks in the latency distribution of LMAN neurons mirrored the inhibitory responses observed in Area X pallidal neurons. Indeed, pallidal neurons displayed inhibitory responses either preceding or following the excitatory component of their response. Inhibition in Area X pallidal neurons, many of which project to the thalamic nucleus DLM (*Fee and Goldberg, 2011*; *Leblois et al., 2009*), induces a fast excitatory response in DLM neurons (*Goldberg et al., 2012*; *Leblois et al., 2009*; *Person and Perkel, 2007*) and thereby activates LMAN through DLM excitatory projections (*Leblois et al., 2009*). The first excitation in LMAN neurons, around 20 ms latency, could therefore be mediated by the fast inhibition observed in pallidal neurons (*Figure 2C*, top panel). Similarly, the slow inhibitory component of pallidal responses to DCN stimulation, with a mean latency around 30 ms (28.2 ± 9.5 ms), likely activates the DLM-LMAN pathway with longer latencies (>50 ms) and may therefore drive the second excitation in LMAN. To confirm that Area X relays the response of LMAN neurons to DCN stimulation, we first blocked glutamatergic transmission in Area X (*Figure 6D*). The response strength was calculated as the total area of the response, containing one or two peaks of excitation when they are present. After application of the glutamatergic blockers to Area X, responses were suppressed in LMAN (*Figure 6E and F*, n = 14 multiunit recordings, response strength decreased from 2.04 ± 0.54 spikes at baseline to 0.89 ± 0.23 spikes following drug injection, paired Wilcoxon test, p=0.001; mean peak excitation decreased from 27.3 ± 7.3 Hz at baseline to 10.9 ± 2.9 Hz following drug injection, paired Wilcoxon test, p=4e-004). Finally, to confirm that the inhibitory components in the pallidal response to DCN stimulation mediate responses in LMAN, we blocked fast GABAergic transmission in Area X with the GABA-A receptor inhibitor gabazine while monitoring the response of LMAN neurons to DCN stimulation (*Figure 6G*). We observed the suppression of LMAN neurons excitatory responses after GABAergic blockade in Area X (*Figure 6H and I*, n = 7 multiunit recordings, response strength decreased from 0.94 ± 0.48 spikes at baseline to 0.06 ± 0.05 spikes following gabazine injection, paired Wilcoxon test, p=0.02; mean peak excitation decreased from 42.51 ± 9.66 Hz at baseline to 9.17 ± 6.68 Hz

following gabazine injection, paired Wilcoxon test, p=0.03). Altogether, our results strongly support the view that DCN inputs are transmitted through the basal ganglia-thalamo-cortical loop via the disinhibition of DLM thalamic neurons by Area X pallidal neurons, evoking an excitatory response in LMAN.

We then tested whether DCN stimulation also drives responses in RA neurons via this loop (*Figure 6J*). DCN stimulation induced strong excitatory responses in RA neurons (*Figure 6K*) with latencies in the 10 to 100 ms range (*Figure 6L*, 30.2 ± 7.8 ms, median: 16 ms), consistent with a transmission through LMAN. Blocking glutamatergic transmission in LMAN significantly reduced the excitatory response to DCN stimulation in RA neurons (*Figure 6K and M*, n = 6 neurons in 5 birds, response strength decreased from 0.8 ± 0.32 spikes at baseline to 0.29 ± 0.12 spikes following drug injection, Wilcoxon test, p=0.009; mean excitation peak decreased from 187 ± 76 Hz at baseline to 71 ± 29 Hz following drug injection, paired Wilcoxon test, p=0.02).

## DCN lesion impairs song learning in juvenile zebra finches

Our experiments provide evidence of a functional disynaptic cerebellum-thalamus-basal ganglia pathway in songbirds. This pathway drives the output nucleus of the basal ganglia-thalamo-cortical loop, LMAN, and drives activity in RA neurons.

As song learning relies on the basal ganglia-thalamo-cortical loop (*Bottjer et al., 1984*; *Brainard and Doupe, 2002*; *Nottebohm et al., 1976*; *Scharff and Nottebohm, 1991*), we tested the hypothesis that the cerebellum contributes to song learning during development. Juvenile zebra finches were subjected to partial lesions in their lateral DCN, either electrolytic (n = 7) or chemical using ibotenic acid (n = 3). Lesions were performed between 55 and 60 days post hatch (56.8 ± 7.5 dph for the lesion group, 57.0 ± 4.5 dph for the sham group), a period which corresponds to the end of the sensory phase of song learning, and to the beginning of the sensorimotor phase (*Figure 7B*). *Figure 7B* displays the spectrograms of the song motifs produced by a tutor and its two fledglings (pupils) after crystallization phase (90 to 100 dph), one of them with a DCN lesion. The pupil that underwent the DCN lesion copied fewer syllables than his control brother. To test for a systematic effect of DCN lesions on song imitation, we compared the quality of tutor imitation of the pupils undergoing partial DCN lesion or sham surgery. To this end, we computed the average imitation score over multiple song bouts (*Mandelblat-Cerf and Fee, 2014*). The song bouts (50–100 in each condition) were carefully sorted among 2 days of recordings before and after the surgery, and after crystallization (90 dph). This was done for birds of both the lesion and sham groups. We found a significant correlation between the proportion of the lateral DCN that was left intact and the relative increase in imitation score between the period preceding the surgery (pre) and the crystallization period (Pearson's correlation coefficient r = 0.7, p=0.03; *Figure 7D*). Moreover, birds with large lesions (<75% lateral DCN left intact, n = 7/10) displayed a lower imitation score than the sham group at crystallization (large lesion group: imitation score of 0.39 ± 0.09, n = 7, sham group: imitation score of 0.51 ± 0.06, n = 6, t-test, df = 11, p=0.04, *Figure 7E, F*). We confirmed this effect of DCN lesions using a custom-written similarity score analysis based on the peak cross-correlation between the spectra of the tutor's motifs and of the pupil's songs (see *Figure 7—figure supplement 3*). In conclusion, partial lesions in the lateral DCN induced a subtle but significant effect on the song acquisition process in juvenile zebra finches, providing evidence that the cerebellum contributes to song learning.

## DCN lesions affect song temporal features in juvenile birds

Imitation scores are affected by both acoustic and temporal features of the song. To understand in more details how the cerebellum may contribute to song learning or production, we compared temporal and acoustic features of the song before and after DCN lesion in juvenile and adult zebra finches.

As exemplified on *Figure 8B*, DCN lesions in juvenile birds induced a consistent drift in syllable duration (*Figure 8B*, see *Figure 8A* and Material and methods for details on how syllable duration is calculated). To determine if and how syllable duration was affected by DCN lesion, we report the relative change in syllable duration induced by the lesion between the baseline (2 days preceding the lesion) and the following period (days 5–6 after lesion, a period chosen to avoid contamination by transient short-term effects of surgery, *Figure 8C*, left panel). Relative changes in syllable duration

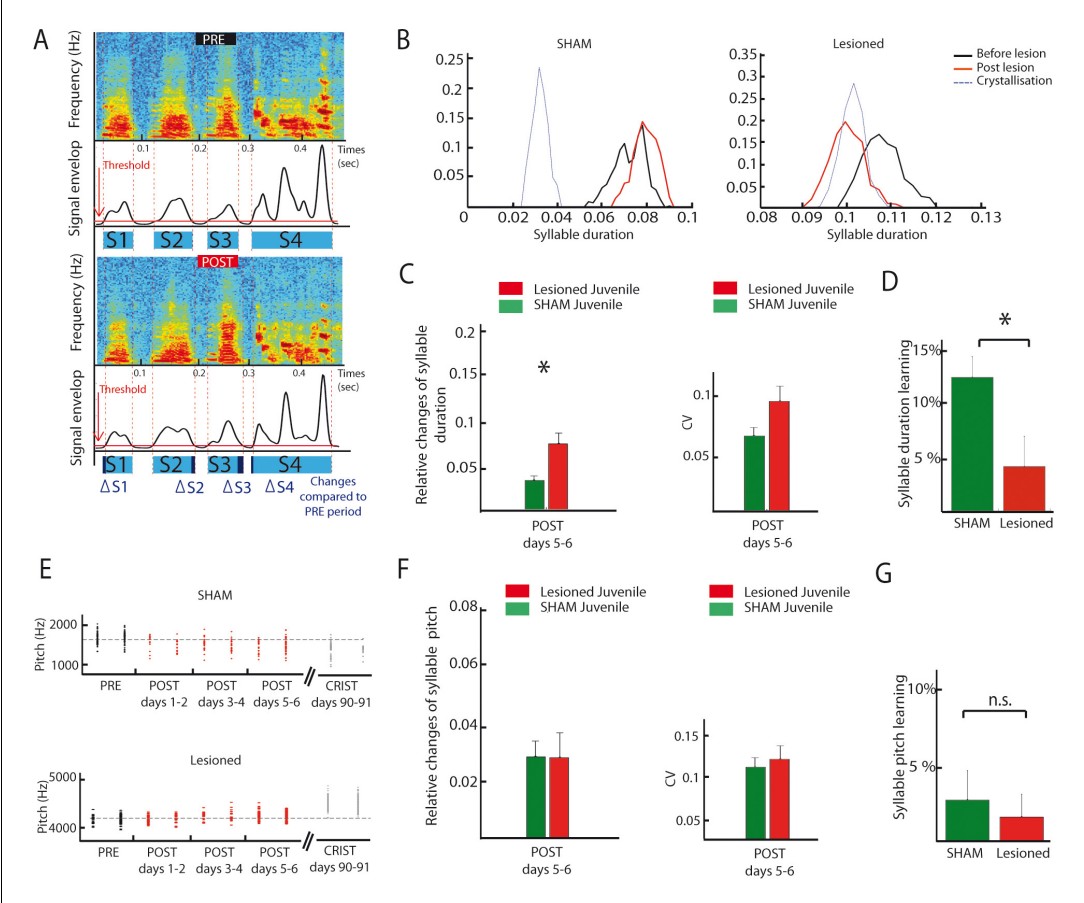

**Figure 8.** DCN lesions effects on syllable duration and fundamental frequency in juvenile zebrafinches. (**A**) Representation of the protocol of syllable duration calculation. The envelop signal of the song was determined, and a threshold was set to determine the beginning and the end of each syllable. (**B**) Distribution of the duration of a syllable over several days in the sham group (left panel) and in the lesion group (right panel, post lesion period in red, crystallization period in grey). (**C**) The duration of syllables before and after the lesion were compared based on their relative changes between these two periods. Left panel: Changes in syllable duration relative to baseline in the sham group (green) and the lesion group (red) for juvenile birds one week after cerebellar lesion. Right panel: CV of the duration of syllables in the sham (green) and lesion (red) juvenile groups one week after cerebellar lesion. (**D**) Learning trajectory for the duration of syllables in the sham (green) and lesion (red) juvenile groups. The learning trajectory is determined by the difference between the relative changes in duration at the crystallization phase (days 90–91) and relative changes in duration during the post days 5–6 after lesion. Learning-related changes in duration in the lesion group significantly differ from those in the sham group (n = 21 syllables in lesion group, mean: 4 ± 2.7%, n = 24 syllables in sham group, mean: 12,1 ± 1,9%, Wilcoxon test, p=0.016). (**E**) Distribution of the fundamental frequency of example harmonic stacks from the sham group (top panel) or the lesion group (bottom panel) over several days (post lesion period in red, crystallization period in grey). (**F**) Left panel: Changes in fundamental frequency relative to baseline for harmonic stacks in the sham (green) and lesion (red) groups. Right panel: CV of the fundamental frequency of harmonic stacks in sham (green) and lesion (red) groups. No difference was observed between different conditions for the fundamental frequency analysis, p>0.05. (**G**) Learning trajectory for the fundamental frequency of harmonic stacks in the sham (green) and lesion (red) groups. The learning trajectories for fundamental frequency were similar in both groups (sham group, n = 10 harmonic stacks, mean: 2.7 ± 1.9%, lesion group, n = 19 harmonic stacks, mean: 1.6 ± 1.5%, Wilcoxon test, p=0.50).
DOI: https://doi.org/10.7554/eLife.32167.016

The following figure supplements are available for figure 8:

**Figure supplement 1.** Effect of cerebellar lesions on the time course of syllable duration, fundamental frequency and amplitude in juvenile birds.
DOI: https://doi.org/10.7554/eLife.32167.017

**Figure supplement 2.** Cerebellar lesions acutely impact syllable duration but do not affect fundamental frequency and amplitude in adult zebra finch song.
DOI: https://doi.org/10.7554/eLife.32167.018

are higher following DCN lesion than in the sham juvenile group (Wilcoxon test, n = 21 syllables in the lesion group, n = 28 syllables in the sham group, p=0.003), demonstrating that DCN lesions impact syllable duration in juvenile birds. In contrast, the variability of syllable duration was not affected by cerebellar lesions (*Figure 8C*, Wilcoxon test, p=0.03, non-significant when correcting for multiple tests, see Materials and methods). In adult birds, the effect of DCN lesions on syllable duration did not reach significance, although a similar trend to increase the relative change in syllable duration compared to sham was observed (*Figure 8—figure supplement 2A–B*, Wilcoxon test, non-significant, see *Supplementary file 1* for detailed statistical value).

These results show that lateral DCN lesions performed at 60 dph do not completely prevent birds from copying a tutor or modifying song syllable duration over development. However, comparing the course of syllable duration of sham and lesion birds between the early sensorimotor phase and the crystallization period suggests that those lesions affect the developmental trajectory of song timing properties (*Figure 8B*). To reveal this, we compared the relative change in syllable duration between the period post 5–6 (after stabilization of acute lesion effects) and 90 dph for the sham lesion groups. *Figure 8D* shows that sham birds display a change of 12 ± 2% during this period, revealing the normal syllable duration learning process at this stage. The group with DCN lesion, on the contrary, displayed a smaller change in syllable duration over the same time interval (4 ± 3%, *Figure 9D*, Wilcoxon test, n = 21 syllables in lesion group, n = 24 syllables in sham group, p=0.02). A closer look at the change in syllable duration after lesion and at crystallization (*Figure 8—figure supplement 1A*) reveals that DCN lesions induce a small acute drift in duration but prevent further changes possibly related to the normal learning process. Thus, lateral DCN lesions performed during the sensorimotor stage impair the learning-related changes in syllable duration.

Our analysis of syllable duration was based on threshold detection (see Materials and methods and *Figure 8A*), and strongly depends on the sound amplitude during singing: a lower sound amplitude, for example, could induce an artifactual decrease in syllable duration in our analysis. We thus checked if DCN lesions affected the amplitude of syllables in adult and juvenile birds (*Figure 8—figure supplement 2A and D*). DCN lesions induced no change in syllable amplitude or in its variability in adults (*Figure 8—figure supplement 2D*, Wilcoxon test, non-significant for all periods) or in juveniles (*Figure 8—figure supplement 1D*, Wilcoxon test, non-significant for all periods, see legend for details), and we can thus rule out any artifactual change in duration due to an effect of the lesion on syllable amplitude.

## DCN lesions did not affect the fundamental frequency of syllables

LMAN, the output nucleus of the song-related basal ganglia-thalamo-cortical loop, is known to drive learning-induced changes in the fundamental frequency of syllables (*Andalman and Fee, 2009*; *Warren et al., 2011*) and to affect its variability (*Kao et al., 2005*). Because we showed that LMAN is under the influence of cerebellar input, lateral DCN lesions could also affect the fundamental frequency of the harmonic stacks in the song motif. Comparison of relative changes in the fundamental frequency of harmonic stacks between the two groups did not reveal any significant change during the early period after lesion (*Figure 8E and F*, n = 19 harmonic stacks for the lesion group and n = 18 stacks for the sham group, Wilcoxon test, p=0.4). We also found no effect of DCN lesion on the learning trajectories of fundamental frequency, measured as the change in frequency between the last period after lesion and the crystallization (*Figure 8G*, sham group, n = 10 fundamental frequency syllable, mean: 2.7 ± 1.9%, lesion group, n = 19 fundamental frequency syllable, mean: 1.6 ± 1.5%, Wilcoxon test, non-significant, p=0.5). Adult birds did not display any significant change in fundamental frequency following DCN lesions either (*Figure 8—figure supplement 2E–F*). Finally, the variability of fundamental frequency was not affected by DCN lesion in adult or juveniles (*Figure 8F*, Wilcoxon test, non-significant, see *Supplementary file 1* for detailed statistical value). Altogether, our results suggest that the cerebellar output from the DCN is not required for the acquisition and adjustment of harmonic stacks fundamental frequency.

## Discussion

Previous investigations into the neural mechanisms of vocal learning in songbirds have focused on the contribution of pallial and basal ganglia circuits (*Mooney, 2009*), ignoring a possible contribution of the cerebellum to avian song learning. Yet, the cerebellum has been proposed to be a crucial

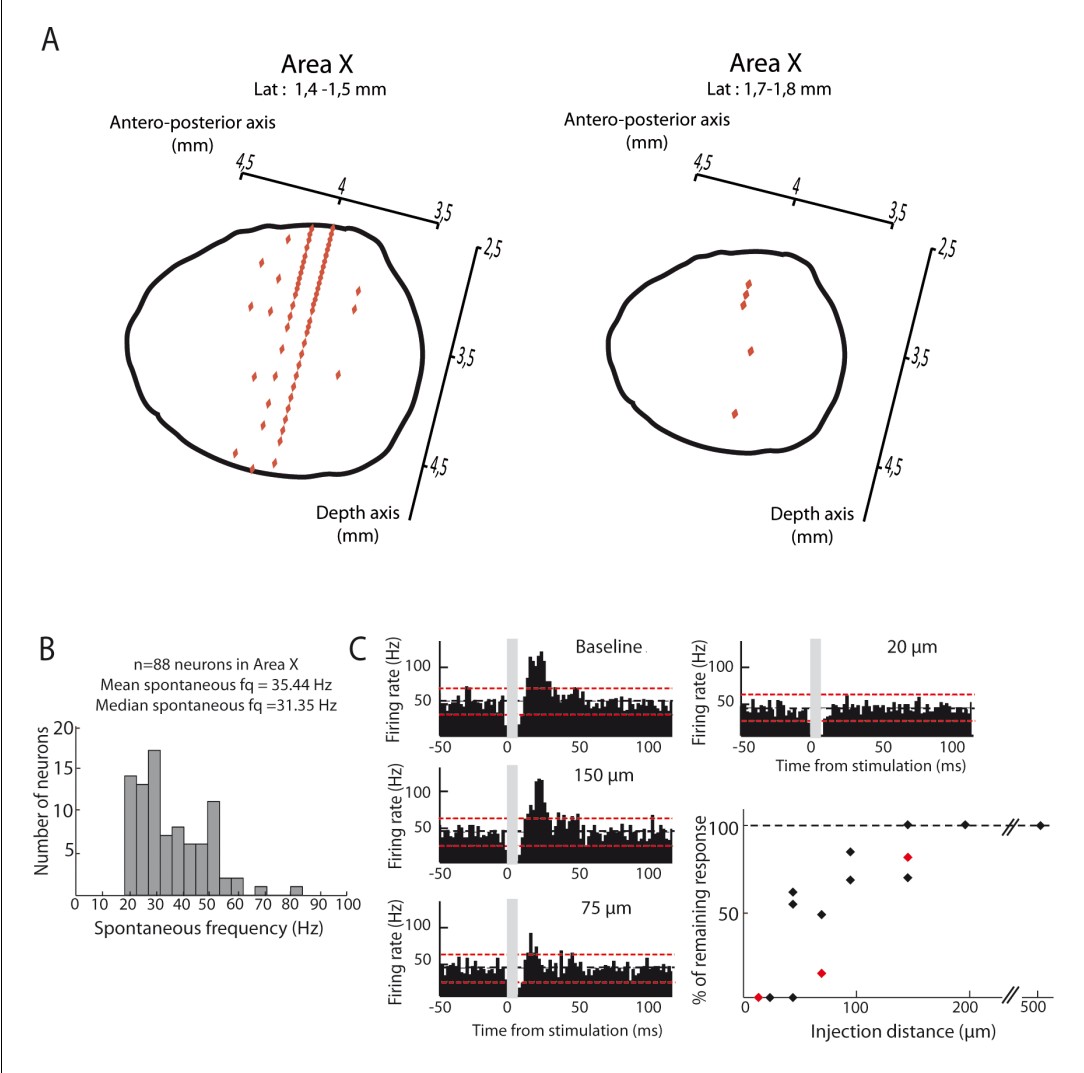

**Figure 9.** Location of Area X neural recordings and effect of the dispersion of pharmacological agents. (**A**) Diagrams showing recordings locations in Area X, for two different lateral plans (1.4–1.5 mm (left panel) or 1.7–1.8 mm (right panel). Each recording point (red diamond) was placed in Area X using antero-posterior and depth coordinates (n = 83 recording sites for a laterality of 1.4–1.5 mm, left panel; n = 5 recording sites for a laterality of 1.7–1.8 mm, right panel). (**B**) Distribution of spontaneous firing rate for neurons recorded in Area X (n = 88 neurons, mean frequency = 35.4 Hz, median frequency = 31.4 Hz). (**C**) Effect of pharmacological blockers (CNQX/APV) on Area X neurons as a function of the distance from the injection site. An example pallidal neuron recorded while injecting the blockers at various distances (baseline: no drug injected): 150, 75 and 20 µm, and the PSTHs displayed show its response to the DCN stimulation after drug injection at the various sites (time bin: 5 ms). The black horizontal dashed line depicts the mean baseline firing rate and red dotted lines indicate confidence intervals (2.5 SD away from the mean baseline firing rate). The population data (bottom right) represent the change in response to DCN stimulation induced by drug injection as compared to baseline for each recorded neuron. Red diamonds correspond to the example shown here.

DOI: https://doi.org/10.7554/eLife.32167.019

element of the speech motor control network in humans. Imaging studies show cerebellar activation during speech production in healthy individuals and patients with cerebellar damage exhibit a variety of speech deficits, the nature of which depends on the location of the lesion. This is not surprising given that the cerebellum is implicated in many, if not all, sensorimotor processes (*Acker-mann, 2008*; *Izawa et al., 2012*) a variety of which are necessary for speech production. Cerebellar lesions impair performance and learning or adaptation of various sensorimotor tasks like pointing, reaching (*Izawa et al., 2012*), timing perception (*Ivry and Spencer, 2013*) and reflex adaptation (*Ito, 1998*). Here, we show that the cerebellum interacts with song-specific circuits in the basal ganglia of songbirds and contributes to the acquisition of song during the development in juvenile birds.

Our data establish a functional excitatory projection from the lateral part of the DCN to the song-related basal ganglia nucleus Area X via a thalamic relay in DTZ in anaesthetized zebra finches. This modulation of basal ganglia activity by the cerebellum then propagates to the cortical target of the song-related basal ganglia loop (LMAN) via the thalamus and is finally conveyed to the premotor nucleus RA. Interestingly, these results are reminiscent of the cerebello-thalamo-basal ganglia pathway recently discovered in mammals (*Bostan et al., 2010*; *Chen et al., 2014*). Thus, our study points the zebra finch as a choice experimental model to investigate the role of the cerebellum and its interaction with the basal ganglia in the learning and plasticity of complex sensory-motor tasks.

## Partial lesions in the cerebellum

The DCN receive strong convergent Purkinje cell inputs from many functional territories in the cerebellar cortex (*Apps and Garwicz, 2005*). To avoid impairing global function or vital sensorimotor abilities (potentially leading to a high post-operative mortality), we limited the extent of our lesions and monitored the animal state and gross motor functions in the days following the lesion. Our quantification of the effect of DCN lesions on song were performed once the transient motor impairments observed following surgery had disappeared and the birds had resumed perching and singing. Gross motor dysfunction was thus unlikely to significantly contribute to the specific changes observed in song. However, only specific lesions of the cerebello-thalamic projections achieved by pathway-specific ablation techniques will rule out this experimental limitation in the future.

## Several types of Area X neurons are potentially involved in the cerebello-thalamo-basal ganglia pathway

Our results indicate that the cerebellar input to the basal ganglia modulates the activity of putative pallidal neurons. We did not directly investigate the response of other neuronal types in this structure. The song-related basal ganglia nucleus, Area X, contains all the neuron types found in the striatum and pallidum in mammals (*Farries and Perkel, 2000*; *Farries and Perkel, 2002*): pallidal neurons, medium spiny neurons and several striatal interneuron types. Only pallidal neurons, however, project outside of the basal ganglia; these share physiological, biochemical and anatomical properties with mammalian pallidal neurons (*Carrillo and Doupe, 2004*). Songbirds pallidal neurons display strong spontaneous activity both in vitro (*Budzillo et al., 2017*; *Farries and Perkel, 2000*; *Farries and Perkel, 2002*) and in vivo (*Person and Perkel, 2007*; *Goldberg and Fee, 2010*) and can therefore be distinguished from the other neuronal populations in the song-related basal ganglia nucleus, the spontaneous activity of which is much lower (*Person and Perkel, 2007*; *Leblois et al., 2009*; *Goldberg and Fee, 2010*). Given the strongly bimodal distribution of spontaneous activity observed in our recording (see Materials and methods) and the relative scarcity of neurons displaying a low spontaneous activity in the song-related basal ganglia nucleus (*Goldberg and Fee, 2010*), our dataset likely contains mostly if not only pallidal neurons. A contribution from a small fraction of spontaneous striatal interneurons cannot, however, be ruled out in the absence of post-hoc histological verification of the recorded cell type.

## Similarities and differences between the cerebello-thalamo-basal ganglia pathways of mammals and songbirds

In mammals, a pathway connecting the cerebellum to the striatum through the thalamus was demonstrated in rodents (*Chen et al., 2014*) and monkeys (*Hoshi et al., 2005*). However, it remains unknown whether and how these cerebellar inputs are conveyed to basal ganglia output neurons and to their thalamo-cortical targets ultimately affecting behavior (*Alexander et al., 1990*). Here, we show in songbirds that the cerebellar signals travel through the basal ganglia-thalamo-cortical circuit and can drive firing in song-related premotor neurons. In monkeys, the dentate nucleus can be divided into two parts: the dorsal part, which has reciprocal projections with motor and premotor cortical areas via the motor thalamus, and the ventral part, which has reciprocal projections with associative and other non-motor cortical areas via non-motor thalamic regions (*Dum and Strick, 2003*; *Kelly and Strick, 2003*; *Orioli and Strick, 1989*). Additionally, anatomical tracing shows that some projections to the thalamus also come from the interpositus and the fastigial nuclei (25%) (*Bostan et al., 2010*; *Hoshi et al., 2005*). In songbirds, our tracing experiments show that one part of the thalamus projects to the song-related basal ganglia nucleus and receives extensive axonal

projections from the most lateral part of the DCN, that could be analogous to the dentate nucleus in mammals (*Arends and Zeigler, 1991*; *Sultan and Glickstein, 2007*; *Voogd and Glickstein, 1998*). However, we found no dorso-ventral contrast in the lateral DCN and we thus make no distinction between potential motor and non-motor parts of this nucleus. Bidirectional tracer injected in the dorsal thalamus revealed a weak, but consistent, projection from the intermediate nucleus, analogous to nucleus interpositus in mammals (*Arends and Zeigler, 1991*; *Sultan and Glickstein, 2007*; *Voogd and Glickstein, 1998*). Although the labeling was less intense in the intermediate nucleus as compared to the lateral one (suggesting weaker projections to the thalamus), both cerebellar nuclei seem to project to the dorsal thalamus, as reported in *Nicholson et al. (2018)*. Both of them may, thereby, be involved in the cerebello-thalamo-basal ganglia pathway studied here.

During our electrophysiological experiments, the stimulation electrode targeted the most lateral part of the DCN, as confirmed histologically. We could observe the activation of the cerebello-thalamo-basal ganglia pathway only with very specific and restrictive placement of the stimulation electrode (see Materials and methods). It is thus unlikely that the responses we report were due to current spread to the neighboring intermediate nucleus. However, the size of the stimulated area can hardly be controlled (*Ranck, 1975*; *Tehovnik et al., 2006*), and we cannot exclude a contribution of the intermediate nucleus to the neural responses we describe here. Further investigations will be necessary to assess this question and determine the role of the putative connections between the intermediate nucleus and the thalamus.

Because striatal and pallidal neurons are intermingled in the song-related basal ganglia nucleus (*Farries and Perkel, 2000*; *Farries and Perkel, 2002*), we could not determine the direct targets of thalamic fibers: - the striatal neurons (as in mammals, *Smith et al., 2004*) - the pallidal neurons - or both. While we focused on the song-related basal ganglia nucleus, the thalamic projections may also reach other parts of the avian basal ganglia. Further investigation using multiple tracing techniques will be necessary to clarify this question and determine which thalamic area projects to which neurons in the basal ganglia.

## Involvement of the cerebellum in timing processing

The cerebellum is a major contributor to timing processes in the brain, both by controlling the duration and variability of movement and by computing the timing prediction necessary to produce an accurate and adapted response during sensorimotor learning. More particularly, clinical observations have highlighted that sensorimotor timing is strongly impaired in patients with unilateral cerebellar lesions. These patients are not able to realize a task in a precise time (*Day et al., 1998*; *Flament and Hore, 1988*; *Izawa et al., 2012*) or to conserve a temporal motor pattern in repetitive and synchronized tasks (*Ivry and Keele, 1989*; *Ivry et al., 2002*). These observations were confirmed with transcranial magnetic stimulation (*Bijsterbosch et al., 2011*; *Théoret et al., 2001*). In repetitive tapping tasks, it has been also shown that motor variability is increased when the lateral cerebellum is inhibited (*Théoret et al., 2001*) and that compensatory mechanisms appear if patients are asked to do bimanual tasks (*Bijsterbosch et al., 2011*; *Franz et al., 1996*; *Théoret et al., 2001*). In mammals, the cerebellum is also responsible for the correct perception of time and time intervals (*Moberget et al., 2008*; *Rao et al., 1997*). Conditioning of the eyeblink reflex, which relies on timing (delay) learning, is impaired following lesions of the cerebellum (*Woodruff-Pak and Thompson, 1985*). In our results, we reveal an involvement of the cerebellum in the duration of syllables but no effects on variability of syllable duration. The relatively small changes in syllable duration induced by DCN lesions may be highly significant behaviorally as zebra finches have been shown to discriminate syllable duration with millisecond precision (*Narula and Hahnloser, 2013*). Knowing which specific features of timing functions (i.e. perception of time or movement timing) is impaired in our songbird model remains an open question.

We revealed a functional connection from the lateral nucleus of the cerebellum to the song-related basal ganglia thalamo-cortical loop, known to generate variability or systematic bias in the fundamental frequency of syllables (*Kao et al., 2005*; *Olveczky et al., 2005*; *Scharff and Nottebohm, 1991*). Thus, a putative role for the cerebellum in the control of fundamental frequency could be expected. No change in fundamental frequency could be detected here either in adults or in juveniles following DCN lesions. Given the relatively small extent of the lesions performed and that other circuits in the song system may compensate for the effect of DCN lesions, we cannot exclude a cerebellar contribution to fundamental frequency.

## Is the cerebello-thalamo-basal ganglia pathway the only functional pathway connecting cerebellum to the song system?

We have revealed a subcortical connection between the cerebellum and the cortico-basal ganglia circuit involved in song learning and plasticity, indirectly affecting activity in the premotor song-related nucleus RA. A more direct connection may also exist from the cerebellum to the motor pathway from HVC to RA that could exert a direct influence on song production. The dorsal thalamus, which mediates cerebellar input to the basal ganglia that we have evidenced here, is also known to project to the pallial nucleus MMAN, which in turn projects to HVC (*Foster et al., 1997*; *Nicholson and Sober, 2015*; *Williams et al., 2012*). This new pathway remains to be characterized by anatomical and electrophysiological experiments to assess the impact of cerebellar input on the cortical pathway during song learning and production. In mammals, the cerebellum is known to project to the motor part of the thalamus, which in turn projects to the motor cortex (*Kelly and Strick, 2003*). This disynaptic connection between the cerebellum and the motor cortex is important in motor control and motor coordination (*Brooks, 1984*) and we therefore hypothesize a contribution of the DCN-DTZ-MMAN-HVC pathway in the production of song in songbirds.

## Potential impact of cerebellar input on basal ganglia

We have shown that a cerebello-thalamo-basal ganglia pathway exists in songbirds, is functional and shares many similarities with the mammalian cerebello-thalamo-basal ganglia pathway. Knowing the role of the cerebellum and the basal ganglia, respectively in supervised and reinforcement learning (*Doya, 2000*), we hypothesize that the cerebellum can participate in basal ganglia functions by sending an error-correction signal related to a detected mismatch between actual and predicted sensory feedbacks. This error correction signal is integrated into the basal ganglia to drive the motor command output during the learning process. As recently reported, the song-related basal ganglia nucleus receives a reward prediction error from the ventral tegmental area that is necessary and sufficient to drive song learning (*Gadagkar et al., 2016*; *Hoffmann et al., 2016*; *Xiao et al., 2018*). The reward prediction error signal from the VTA and the cerebellar error correction signal could cooperate within the basal ganglia to achieve faster and more efficient sensorimotor learning. In this context, the cerebellar input could modulate plasticity of the avian equivalent of the cortico-striatal connections, as described in mice (*Chen et al., 2014*), and thereby regulate the learning rate in the basal ganglia circuits.

In songbirds, the basal ganglia-thalamo-cortical loop is necessary for song learning and plasticity (*Brainard and Doupe, 2002*; *Olveczky et al., 2005*). Our data suggest that these functions - presently attributed to the basal ganglia-thalamo-cortical loop only - may also be influenced by the cerebellum through its subcortical connection to the song-related basal ganglia nucleus.

Finally, the subcortical pathway from the cerebellum to the basal ganglia is involved in dystonia (*Calderon et al., 2011*; *Fremont et al., 2017*; *Neychev et al., 2008*; *Tewari et al., 2017*). The existence of the cerebello-thalamo-basal ganglia pathway makes the songbird model, classically used as a model to study vocal learning, a good model for further investigations of the cooperation between cerebellum and basal ganglia in sensorimotor learning and its dysfunction in movement disorders.

## Materials and methods

### Animals

All the experiments were performed in adult male zebra finches (*Taeniopygia guttata*), >90 days post-hatch unless otherwise specified. Birds were either reared in our breeding facility or provided by a local supplier (Oisellerie du Temple, L'Isle d'Abeau, France). All animals had constant access to seeds, crushed oyster shells and water. Seeds supplemented with fresh food and water were provided daily. Birds were housed on a natural photoperiod (both in the aviary and in sound isolation boxes during the behavioral experiment). Animal care and experiments were carried out in accordance with the European directives (2010–63-UE) and the French guidelines (project 02260.01, Ministère de l'Agriculture et de la Forêt). Experiments were approved by *Paris Descartes University* ethics committee (Permit Number: 13–092).

## Surgery

Before surgery, birds were first food-deprived for 20–30 min, and an analgesic was administered just before starting the surgery (meloxicam, 5 mg/kg). The anesthesia was then induced with a mixture of oxygen and 3–5% isoflurane for 5 min. Birds were then moved to the stereotaxic apparatus and maintained under anesthesia with 1% isoflurane. Xylocaine (31.33 mg/mL) was applied under the skin before opening the scalp. Small craniotomies were made above the midline reference point, the bifurcation of the midsagittal sinus, and above the structures of interest. Stereotaxic zero in antero-posterior and mediolateral axis was determined by the sinus junction. To ease the access to the cerebellum, we used a head angle of 50°. The stereotaxic coordinates used for each brain structure are summed up in *Table 1*.

## Anatomical tracing

We performed iontophoretic injections of fluorescent dye using dextran conjugates with Alexa 594 and Alexa 488 (Thermofischer, 5% in PBS 0.1M 0.9% saline) in targeted cerebral structures (lateral DCN and Area X nucleus) using a glass pipette with a small (10 μm) tip and ±5 μA DC pulses of 10 s duration, 50% duty cycle, applied for 3 min. In the cerebellum, to be sure that the injection was constrained to the lateral deep cerebellar nucleus, we verified that the retrograde labeling of Purkinje cells was limited the most lateral sagittal zone (*Figure 1D*, and *Figure 1—figure supplement 1*).

In additional tracing experiments, 250 nL of cholera toxin tracers coupled with Alexa 488 (Thermofischer, diluted in PBS 0.1M 0.9% saline) were pressure-injected with a Hamilton syringe (1 μL, Phymep, Paris, France), at 100 nL per minute, at each injection site (two injection sites per brain hemisphere). Birds were then housed individually for three days after injection to allow for dye transport.

## In vivo electrophysiology

Recordings in Area X, LMAN, and RA were made with a tungsten electrode with epoxy isolation (FHC, impedance varying from 3.0 to 8.0 MΩ depending on the type of neuron recorded). Acquisition of the signal was done with the AlphaOmega software, using low-pass (frequencies below 8036 Hz) and high-pass (frequencies above 268 Hz) filters to only detect the spike signal. The sampling frequency was 22,320 Hz. In Area X, the recorded neurons displayed a bimodal distribution of spontaneous firing rate, above 25 Hz or under 10 Hz. We considered neurons with frequency above 25 Hz as pallidal neurons in Area X (*Leblois et al., 2009*; *Person and Perkel, 2007*). Other neurons in Area X with spontaneous firing rates under 10 Hz were not taken into account in the present study. Distribution of the pallidal-like neurons firing rate is represented in *Figure 9C*. Note that the level of spontaneous activity is different under anesthesia compared to what was seen in awake birds

**Table 1.** Stereotaxic coordinates summary.

Head and arm angle (on the mediolateral axis) are expressed in degrees, anteroposterior and mediolateral coordinates are expressed in millimeters from the sinus junction, and depth coordinates in millimeters from the surface of the brain. DCN: deep cerebellar nuclei. LMAN: lateral magnocellular nucleus of the nidopallium, MMAN: medial magnocellular nucleus of the nidopallium, HVC: used as a proper name, DTZ: dorsal thalamic zone.

| Structure | Head angle (°) | Arm angle (°) | Antero-post (mm) | Medio-lateral (mm) | Depth (mm) |
|---|---|---|---|---|---|
| Area X | 50 | 0 | 4.0 | 1.5 | 3.0–4.0 |
| | 50 | 15 | 4.0 | 2.7 | 3.5–4.5 |
| DCN | 50 | 15 | −2 | 2.5 | 3.5 |
| | 50 | 0 | −1.5/−1.8/−2.1 | 1.3 | 3.4 |
| DTZ | 50 | 0 | −0.3 | 1.2 | 4.3–4.5 |
| LMAN | 50 | 0 | 4.1 | 1.8 | 2.3–2.5 |
| | 50 | 15 | 4.1 | 3.0 | 2.4–2.6 |
| MMAN | 50 | 0 | 4.1 | 0.5 | 2.3–2.5 |
| | 50 | 15 | 4.1 | 1.7 | 2.4–2.6 |

DOI: https://doi.org/10.7554/eLife.32167.020

(*Goldberg and Fee, 2010*) and can vary depending on the specific drug used (*Brooks, 1984*). This may explain the slight difference in spontaneous activity among neurons recorded here as pallidal, compared to previous studies performed under urethane anesthesia (*Leblois et al., 2009*; *Person and Perkel, 2007*), known to preserve awake-like cortical activity (*Albrecht et al., 1990*).

A single-pulse electrical stimulation in the lateral deep cerebellar nucleus (DCN) was applied through a bipolar electrode during recording of different structures in the contralateral basal ganglia nucleus (Area X), the lateral part of the magnocellular nucleus (LMAN), the medial part of the magnocellular nucleus (MMAN), and robust archopallium nucleus (RA). The duration of the stimulation was 1 ms, with an inter-stimulation time of 1.6 s, and the intensity ranged from 0.1 to 4 mA. Despite long stimulation duration, observed responses in recorded neurons were stable over time. We aimed to place the stimulation electrode within the lateral cerebellar nucleus, and the positioning of the electrode was confirmed histologically (see next paragraph). However, we cannot completely rule out that the stimulation current did spread to the nearby interpositus nucleus. Other possible confounds due to non-specific effects of stimulation could be that brainstem structures that communicate with the forebrain song system, and fibers of passage that descend from RA to the brainstem could be activated. However, such non-specific effects are highly unlikely due to the distance between the DCN and the song-related brainstem structure (>1 mm), and their separation by the fourth ventricule. Most importantly, a small offset in the placement of the stimulating electrode most often led to the total disappearance of the responses evoked in the basal ganglia circuit, and it is thus unlikely that neurons or fibers away from the stimulation site are mediating the observed responses.

## Pharmacology

During electrophysiological experiments, drugs were applied locally by pressure with small tip glass pipette (10 μm) and nitrogen picospritzer (Phymep, Paris, France) during 5 ms. The volumes injected are around 100–200 nL, with a maximal total injected volume during one experiment of 500 nL. We used a mix of NBQX 5 mM (Sigma Aldrich, diluted in PBS 0.1M 0.9% saline) and APV 1 mM (Sigma Aldrich, diluted in PBS 0.1M 0.9% saline) to block glutamate receptors. Except for Area X blocking, that requires several coordinates injection in order to block a large part of this structure, all blockade are made in one location with two puff injections.

To determine the drug dispersion, we injected NBQX/APV at several distance from the recorded neuron in Area X. We then compared neurons responses strength (see Data analysis for the quantification protocol) with and without drug injection to assess the percentage of resting response (*Figure 9C*). Drug dispersion experiments indicate that excitatory responses were not impacted if the distance between the recorded neuron and the drug injection was more than 200 μm (n = 3 neurons for 200 μm, mean resting response: 94,3 ± 9,6%). For distances between 150 and 50 μm, we observe a progressive decrease in excitatory responses, with a halving of excitatory responses for distances around 75 μm (n = 3 neurons for 150 μm, mean resting response: 83,3 ± 15,5%; n = 2 neurons for 100 μm, mean resting response: 76,1 ± 11,5%; n = 2 neurons for 75 μm, mean resting response: 37,1% ± 33,2; n = 3 neurons for 50 μm, mean resting response: 51 ± 29,6%). Then, excitatory responses in pallidal neurons were totally prevented if the distance between the recorded neuron and the drug injection was less than 50 μm (n = 0.5 neurons, mean resting responses: 0%).

Moreover, glutamatergic blockade effect on the recorded neuron firing rate was quantified (*Figure 10*, see Data analysis for the quantification protocol) during baseline, drug injection and washout conditions. No significant effect of the drug injection on the firing rate of recorded neurons was observed (*Figure 10A-E*, paired Wilcoxon test, see Legend for p values), except for recordings in RA during LMAN glutamatergic blockade (*Figure 10F*, paired Wilcoxon test, p=0.0313).

During LMAN recordings, we also blocked inhibition transmission in Area X. To do so, we used gabazine 1 mM (Sigma, diluted in PBS 0.1M 0.9% saline).

## Data analysis

Analyses of recorded neurons after DCN stimulation were done using Spike2 and Matlab. Spike sorting was performed with the software Spike2 (CED, UK), using principal components analysis of spike waveforms. For Area X neurons, and RA neurons, we managed to record single units, and we focus on these single unit neurons in the analysis. In the LMAN and MMAN nuclei, we chose to record

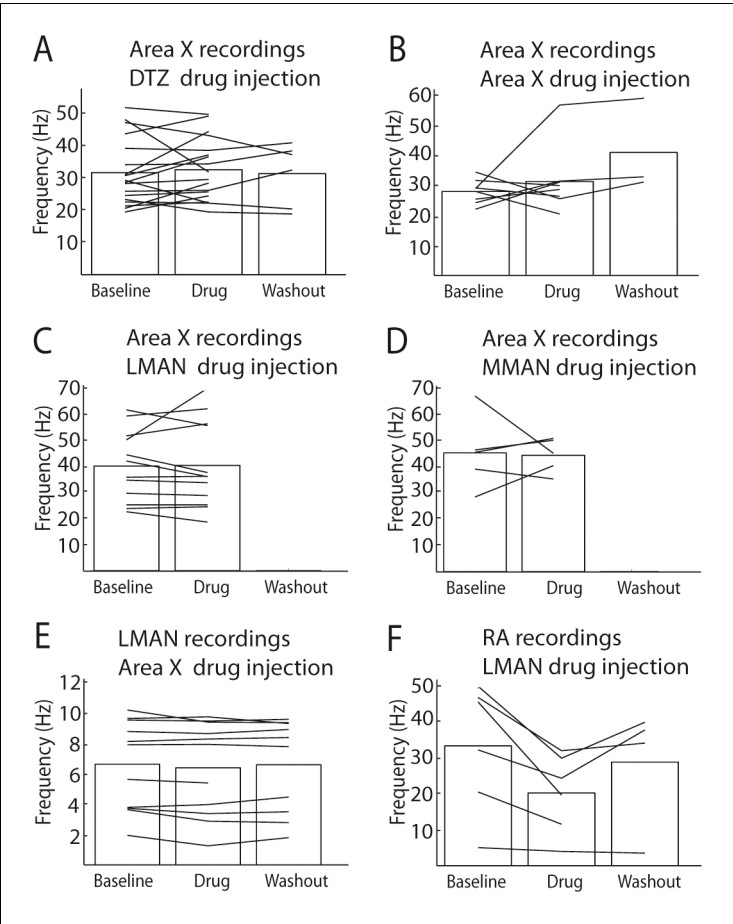

**Figure 10.** Effect of drug injections (NBQX/APV) on spontaneous activity. (**A**) Effect of NBQX/APV injections in DTZ on the spontaneous activity of Area X pallidal neurons. No difference was observed between baseline and drug conditions (n = 16 neurons in 8 birds, paired Wilcoxon-test, p=0.4). (**B**) Effect of NBQX/APV injections on the spontaneous discharge of Area X pallidal neurons. No significant differences were observed in Area X pallidal neurons discharge between baseline and drug conditions (n = 8 neurons in 7 birds, paired Wilcoxon-test, non-significant, p=0.6). (**C**) Effect of NBQX/APV injections in LMAN on Area X pallidal neurons spontaneous discharge. No difference was observed between baseline and drug conditions (n = 12 neurons in 6 birds, paired Wilcoxon-test, p=0.6). (**D**) Effect of NBQX/APV injections in MMAN on Area X pallidal neurons spontaneous activity. No difference was observed between baseline and drug conditions (n = 5 neurons in 2 birds, paired Wilcoxon-test, p=1). (**E**) Effect of NBQX/APV injections in Area X on LMAN neurons spontaneous discharge. No difference was observed baseline and drug conditions (n = 14 multiunit recordings in 5 birds, paired Wilcoxon-test, p=0.1). (**F**) Effect of NBQX/APV injections in LMAN on RA neurons spontaneous activity. RA neurons activity decreased significantly following glutamatergic transmission blockade in LMAN (n = 6 neurons in 5 birds, paired Wilcoxon-test, p=0.03).

DOI: https://doi.org/10.7554/eLife.32167.021

mostly multiunit activity. Indeed, most neurons in these nuclei exhibit very low spontaneous activity (~1 sp/s), leading to wide fluctuation in the PSTH estimate of baseline activity preceding stimulation with high temporal resolution (time bin: 10 ms) and making it difficult to estimate response latency, strength and duration. Instead multi-unit activity with higher baseline levels allows better baseline statistics and narrower confidence intervals for the detection of the response to stimulation.

Spike train analysis was then performed using Matlab (MathWorks, Natick, MA, USA). We calculated peri-stimulus time histograms (PSTH) of recorded neurons after DCN stimulation. PSTHs were calculated with a 2 ms bin for neurons in Area X and RA. For structures with low firing rate (LMAN and MMAN) the time bin was 10 ms to limit bin-to-bin fluctuations in spike count. We calculated the mean and the standard deviation (SD) of the firing rate over the period preceding the stimulation

(50 ms for Area X and RA, 100 ms for LMAN and MMAN), and we considered that a neuron exhibited a significant response to the stimulation when at least two consecutive bins of the PSTH were above (for excitation) or below (for inhibition) the spontaneous mean firing rate $\pm 2.5*SD$. The return of two consecutive bins at the spontaneous mean firing rate $\pm 2.5*SD$ indicated the end of the response. We defined the latency of response as the time between the stimulation onset and the beginning of the first excitatory or inhibitory response. Response strength was calculated as the sum of the difference between the PSTH values and the mean baseline firing rate over the entire response period and represents the average number of excess (default) spikes induced by a single stimulation. For neurons in Area X and RA, the response strength was calculated over the first peak of excitation only (as most responses did not elicit two peaks of excitation, see Results). For LMAN and MMAN neurons recording, neurons tended to display bimodal responses (see Results) and both the first and second excitation peaks were considered to calculate the response strength. We also report the peak firing rate in the response period as the maximal value of the PSTH. The PSTHs are displayed either as histograms or as solid curves with gray shaded area surrounding the curve representing the SD of the baseline firing rate.

## Lesion experiments

Lesions were performed in the DCN of juvenile zebra finches. We targeted the most lateral DCN, analogous to the dentate nucleus in mammals. In a first group of birds (n = 7), a partial electrolytic lesion was performed in the lateral deep cerebellar nucleus by passing 0.05mA during 30 s through a tungsten electrode. Lesions were made at three points (see the stereotaxic coordinates in *Table 1*, DCN coordinates, second row). In a second experimental group (n = 3), chemical partial lesion was performed using ibotenic acid in 1 μL Hamilton syringe, with a rate of 100 nL/min. We also performed injections at three locations (see *Table 1*, DCN coordinates) injecting 150 nL per point. Sham lesions were performed in another group of age-matched juvenile birds. Sham birds underwent the same surgery as the lesion group, with a stimulating electrode placed at the lesion location but no current was applied. Both lesion and sham protocols were done around 57 days post hatch (56,8 ± 7,5 days post hatch for lesion group, 57.0 ± 4,5 days post hatch for sham group). Following surgery, the behavior of birds was closely monitored for a few days to ensure proper recovery. Many birds underwent temporary motor deficits (postural and balance troubles) for a couple of days but recovered very quickly and were all perching and feeding normally 48 hr after surgery. Singing usually resumed after 48 hr, or at most after 72 hr. Each juvenile (sham and lesion) was put in a recording box one week before the lesion experiment, and recorded using Sound Analysis Pro software (SAP, *Tchernichovski et al., 2001*). To prevent any deficit due to the lack of tutor, we presented the tutor to the juvenile two hours per day until the bird underwent the surgery. All birds had same access to their respective tutors. After the surgery, each juvenile was recorded until the crystallization phase (30 days after the surgery experiment).

## Histology

For the anatomical tracing protocol: Birds were sacrificed with a lethal intraperitoneal injection of pentobarbital (Nembutal, 54.7 mg/mL), perfused intracardially with PBS 0.01M followed by 4% paraformaldehyde as fixative. The brain was removed, post-fixed in 4% for 24 hr, and cryoprotected in 30% sucrose. We then cut 40 μm thick sections in the parasagittal plane with a freezing microtome. Slices were mounted with Mowiol (Sigma Aldrich) and observed under an epifluorescence (Leica Microsystems, Leica DM 1000, Nanterre, France) or a confocal microscope (Zeiss, LSM 710, France). Images were analyzed using ImageJ software (Rasband WS, NIH, Bethesda, Maryland, USA).

After electrophysiological recordings, the bird was perfused as described above. Then, brain was removed, post-fixed one day in PFA 4%, store in sucrose 30%, and we did 60 μm slices with Nissl staining to control the stimulation electrode and recording electrode tracts.

For the lesion protocol: All juvenile birds were sacrificed at 100 dph using the protocol previously described for tracing protocol. We then cut 60 μm-thick cerebellar sections in the horizontal plane with a freezing microtome. We did Nissl staining to check lesions locations. Slices were mounted with Mowiol (Sigma Aldrich) and observed under a transmitted-light microscope (Leica Microsystems, Leica DM1000, Nanterre, France). With ImageJ software (Rasband WS, NIH, Bethesda,

Maryland, USA), we calculated the area of lesion for each nucleus compared to the control nucleus in the other hemisphere.

## Song imitation analysis

Songs were continuously recorded using Sound Analysis Pro software (SAP, *Tchernichovski et al., 2001*). Songs were then sorted and analyzed using custom Matlab programs (https://github.com/aleblois/Pidoux_et_al_2018.git, *Pidoux and Leblois, 2018*; MathWorks, Natick, MA, USA; copy archived at https://github.com/elifesciences-publications/Pidoux_et_al_2018). Briefly, the program detected putative motifs based on peaks in the cross-correlation between the sound envelope of the recorded sound file and a clean preselected motif. Putative motifs were then sorted based on their spectral similarity with the pre-selected clean motif, using thresholds set by the experimenter. Song bouts including one or more song motifs separated by less than 500 ms of silence were then cut based on the same sound amplitude threshold. This analysis allowed us to successfully sort >98% of the songs produced by a bird on a given day (assessed by comparing hand sorting with the automated sorting by the program). We calculated the spectrogram of extracted song through fast Fourier transforms using 256-point Hanning windows moved in 128-point steps. Among all songs produced by a juvenile in each considered condition: before and after lesion, as well as at crystallization, 50 to 100 song clean song bouts with no noise contamination (cage noise, wing flaps, . . .) were randomly-selected songs to be compared to the tutor's selected motifs using the procedure described in *Mandelblat-Cerf and Fee (2014)*. The corresponding Matlab program provides 3 outputs: an acoustic similarity index and a sequencing similarity index, which are compiled together into a single imitation score. We only reported the final imitation score in the present study as the relatively mild effect of DCN lesion did not allow to distinguish acoustic and sequencing effects. A custom-based analysis relying on the cross-correlation between spectrograms was also applied (see *Figure 8—figure supplement 2* and its legend for method) to confirm the default in imitation revealed by the imitation score.

## Song temporal features, fundamental frequency and amplitude analysis

For each bird undergoing DCN lesion, or sham-lesion experiments, spectrograms of 500 randomly-selected, manually-checked renditions of the stereotyped motif were stored. To determine the acute effect of the lesion, we analyzed several song features in the first week after the lesion, grouped values for two consecutive days and named these periods pre, post days 1–2, post days 3–4, post days 5–6. Moreover, the same analysis was performed at days 90–91 (after crystallization), to determine the learning trajectory of each song feature. For each considered day, roughly 500 motifs were used to calculate the duration, fundamental frequency and amplitude of each syllable using the following procedure. The sound envelop was generated, and a threshold was determined, corresponding to the lowest envelop signal value (i.e. the smallest amplitude in the motif). The beginning and the end of each syllable was determined as the time at which the song envelop crossed this threshold. This process was performed for each syllable type in the motif (generally 4 to 6 syllables per motif), on our spectrograms of 500 randomly-selected motifs. To pool the data from all syllable types, we normalized syllable duration by doing the absolute ratio between the syllable duration in the post periods (post days 1–2, post days 3–4, post days 5–6, crystallization) over the duration syllable calculated in the period before lesion. This calculation reveals the relative duration changes compared to the pre-lesion period, i.e. how the duration evolved over the time. The variability of syllable duration between syllable types from a given bird and between birds were not significantly different (Kruskal and Wallis test, n = 7 birds/21 syllables for lesion group, p=0.69 between birds and p=0.56 between syllable, and n = 10 birds/27 syllables, p=0.75 between birds and p=0.71 between syllables), allowing to compare all syllable types from a given group (sham vs lesion) in each condition. Relative syllable amplitude was determined as the peak sound envelop during the syllable divided by the peak sound envelop over the whole motif. The syllable fundamental frequency was determined for each syllable type displaying a clear harmonic structure based on peaks in the autocorrelation function, as in the study by *Kao and Brainard (2006)*. For some syllables, several sub-syllabic elements had a clear and distinct fundamental frequency, leading to several fundamental frequency measurements in the same syllable. Normalizations, identical to the one described for syllable duration, were applied for the amplitude and fundamental frequency of all syllable types. Finally, the

learning trajectory was calculated for each group and each feature using the relative change at crystallization minus the relative change values for post days 5–6.

## Statistics

Numerical values are given as mean ± SD, unless stated otherwise.

### Electrophysiology

As the goal of pharmacological experiments was to look at the effect of glutamatergic transmission blockade on baseline response strength induced by DCN stimulation, we compared the mean response strength during two conditions: the baseline condition and the drug condition. To do so we performed a paired Wilcoxon test between the control response and that after application of drugs. We used non-parametric statistical tests because of the small number of neurons recorded (less than 30 neurons in each experiment).

### Behavior

Given our initial hypothesis that the cerebellum may contribute to song learning, we planned to compare the similarity between juvenile and tutor songs at crystallization (90 dph) in the lesion and sham groups. The similarity scores in these two groups were compared using a paired Wilcoxon test (MathWorks, Natick, MA, USA). Additionally, we tested whether there was a significant correlation between the size of the lesion and the improvement in tutor song imitation after surgery. To this end, we calculated the correlation coefficient between the lesion size (proportion of DCN left unaffected, determined histologically for DCN lesion birds, and assigned to 100% for sham-lesion birds) and the normalized song similarity at crystallization (similarity at 90 days post hatch/similarity before surgery). We tested the hypothesis of no correlation: each p value was determined as the probability of obtaining a correlation larger than the observed value by chance, when the true correlation is zero (MathWorks, Natick, MA, USA).

For duration, fundamental frequency and amplitude relative changes, our goal was to compare the relative changes between sham and lesion groups one week after the lesion (days 5–6, to avoid short-term effects of surgery). To do so, we used Wilcoxon test (MathWorks, Natick, MA, USA) to compare values in the sham group to the values in the lesion group at days 5 and 6 after cerebellar lesion. We used non-parametric statistical test because of the non-normal distribution of values. CV quantification follows the same statistical procedure. A summary of statistical values is provided in *Supplementary file 1*.

## Acknowledgements

We are grateful to Claude Meunier, David Hansel and David J Perkel for their comments on the manuscript. This work was supported by the Agence Nationale pour la Recherche (ANR, program 'Retour Post-Doc', Grant number ANR-10-PDOC-0016) and by the city of Paris, France (program 'Emergence', Grant number DDEEES 2014–166).

## Additional information

### Funding

| Funder | Grant reference number | Author |
|---|---|---|
| Agence Nationale de la Recherche | ANR-10-PDOC-0016 | Arthur Leblois |
| City of Paris, Emergence Program | DDEEES 2014–166 | Arthur Leblois |

The funders had no role in study design, data collection and interpretation, or the decision to submit the work for publication.

## Author contributions
Ludivine Pidoux, Software, Formal analysis, Investigation, Visualization, Methodology, Writing—original draft, Writing—review and editing; Pascale Le Blanc, Resources, Methodology; Carole Levenes, Conceptualization, Writing—review and editing; Arthur Leblois, Conceptualization, Supervision, Funding acquisition, Validation, Writing—original draft, Project administration, Writing—review and editing

## Author ORCIDs
Ludivine Pidoux (iD) http://orcid.org/0000-0002-5268-4067

## Ethics
Animal experimentation: Animal care and experiments were carried out in accordance with the European directives (2010-63-UE) and the French guidelines (project 02260.01, Ministère de l'Agriculture et de la Forêt). Experiments were approved by Paris Descartes University ethics committee (Permit Number: 13-092).

## Decision letter and Author response
Decision letter https://doi.org/10.7554/eLife.32167.025
Author response https://doi.org/10.7554/eLife.32167.026

# Additional files

## Supplementary files
• Supplementary file 1. Related to *Figure 8*. Statistical values for Wilcoxon test with Bonferonni correction. For each period in each group (adults or juveniles and sham or lesioned birds) and each features (duration, fundamental frequency and amplitude) number of birds, number of syllables, mean, median, standard deviation and SEM were reported. p values for Wilcoxon test with Bonferonni correction were calculated for each repeated test. N.S.: non-significant.
DOI: https://doi.org/10.7554/eLife.32167.022

• Transparent reporting form
DOI: https://doi.org/10.7554/eLife.32167.023

## Data availability
All data generated or analysed during this study are included in the manuscript and supporting files.

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
