## [Decision Letter]

Thank you for submitting your article "A subcortical circuit linking the cerebellum to the basal ganglia engaged in vocal learning" for consideration by *eLife*. Your article has been reviewed by 4 peer reviewers, including Jennifer L Raymond as the Reviewing Editor and Reviewer #4, and the evaluation has been overseen by Andrew King as the Senior Editor. The following individuals involved in review of your submission have agreed to reveal their identity: Richard Mooney (Reviewer #2).

The reviewers have discussed the reviews with one another and the Reviewing Editor has drafted this letter summarizing their concerns. Please provide details on how you could proceed to address these issues and provide an estimate of the time it would take to do so. The Editorial Board and reviewers will consider your responses and offer a binding recommendation on the outcome of this submission as soon as possible.

Summary:

Birdsong, as with speech but unlike most other animal vocalizations, is a learned behavior. Because vocal learning is a rare trait in the animal world, the songbird has become one of the favorite models for exploring the neurobiological basis of vocal learning. A wealth of evidence implicates cortico-basal ganglia (BG) circuitry in birdsong learning. One remarkable gap in our understanding is whether the cerebellum also plays a role in birdsong learning. Anatomical and physiological studies in a range of species have revealed prominent connections between the cerebellum and basal ganglia via the thalamus, suggesting a role for cerebellar processing in behaviors that rely on basal ganglia-thalamo-cortical pathways. A cerebellar role in birdsong might be expected given the importance of the cerebellum to human speech and the more general involvement of the cerebellum in learning rapid and precisely timed behaviors. Remarkably, evidence of a functional interaction of the cerebellum with known song circuitry has been lacking and the role of the cerebellum in song learning has remained unexplored. The current study by Pidoux et al., provides the first evidence that the cerebellum functionally interacts with basal ganglia circuitry important to song learning in birds and that cerebellar outputs from the deep cerebellar nuclei (DCN) may contribute to song learning. The authors replicate an earlier study showing a pathway from DCN to Area X through thalamic nucleus DTZ. They then show responses in various regions of the song system to DCN stimulation and report that these responses require signal transmission through DTZ. Finally, they claim that DCN inactivations impair song learning. The birdsong field has long wondered about possible involvement of the cerebellum and demonstrating such involvement would be a provocative and high-impact result, providing a blueprint for integrating the cerebellar circuit into the well-established network of brain areas already known to contribute to vocal learning.

Essential revisions:

1) All four reviewers agreed that the behavioral studies are where the greatest potential significance of the study lies and also where the experimental evidence, as presented, falls farther from the mark. Only 10 random songs were analyzed pre- and post-lesion and are summarized with a single similarity score. More extensive analysis of the behavioral results is crucial.

1a) The authors lesion the DCN in juvenile birds using either chemical or electrolytic methods. They report that learning (copying?) of a tutor song is impaired by these lesions. Several problems arise, though, in part because they employ a novel analysis method in place of rather than in addition to acoustic analysis methods that are standard in the field (i.e., SAP). It is unclear how they calculated similarity score (is it the peak of the cross-correlation, as suggested in subsection “DCN lesion impairs song learning in juvenile zebra finches”, or is it really the sum of the cross-correlation as shown in subsection “Song analysis”? In either case, if the score is based on the simple normalized cross correlation, it would be difficult to interpret their data because the score can be very low when juveniles improvise the gap/syllable durations or syllable orders, which often happens in normal zebra finches. A parallel analysis of song data using more established methods and showing more examples of spectrograms of the adult songs of birds with juvenile DCN lesions would be necessary for reviewers and other readers to assess the significance of the behavioral outcomes. The authors should also show the raw similarity score at 60 dph because there should be large variability in learning already at the age, which could significantly affect the normalized similarity score at 90 dph.

1b) Is there a specific contribution of this cerebellar circuit to vocal learning? In the final figure, the authors demonstrate that lesions of the cerebellar dentate nucleus impair the ability of juvenile birds to copy their tutor's song using a similarity score metric. It is unclear, however, whether there are any specific differences in the learned song resulting from cerebellar lesion that would not be seen by lesioning other brain areas involved in song learning. The manuscript would be greatly enhanced if the authors could perform further analyses designed to identify whether any discrete song features are selectively impaired by cerebellar lesions. For example, is there any specific temporal or pitch-related deficit in song production following tutoring? Is the trajectory of learning altered? Is there any evidence based on learning trajectory that learning is simply slowed, and thus incomplete at the time of measurement? Such information would be extremely valuable in order to compare these experiments with the learning deficits that result from lesions to other brain areas involved in vocal learning and could help guide meaningful hypotheses about the specific role of cerebellum in vocal learning.

1c) The description of the timing of the DCN lesions is critical to interpreting what type of learning – sensory versus sensorimotor – the DCN may be contributing to. The timeline in Figure 7 suggests that DCN lesions may be made late enough in development to largely affect song copying rather than tutor song memorization. Language in the Results section and Discussion section should be precise when referring to song copying versus sensory learning. The authors should discuss whether their experimental design allows them to assign the effects of DCN lesions more exclusively to song copying. This is a place where more description of the experimental design in the results would help a general audience.

1d) The central conclusion of the paper – that DCN is involved in song learning – rests on comparing DCN-lesioned and sham-lesioned birds. The authors claim that these two groups are different because sham-lesioned birds imitate the tutor with a p value of 0.04, while DCN-lesioned birds "don't imitate" with a p value of 0.06. This kind of egregious misuse of statistics is the reason many results published in the literature cannot be replicated! Comparing p values of two groups is simply not valid; for instance, these values could be different because of differences in sample sizes or differences in sample variances. The only way to compare two conditions is by comparing the data directly in a two-sided statistical test. Sorting the raw values in Figure 7F makes it clear that the two populations are not actually statistically different. Given the high quality of imitation in some of the DCN-lesioned birds, it appears that the opposite conclusion may be correct – that DCN is not necessary for learning.

1e) It is inadequate to state in subsection “DCN lesion impairs song learning in juvenile zebra finches” that DCN lesions in adults had no effects on song and then not show any supporting data. Finding that the DCN only affects learning or also affects adult performance might be equally interesting. But not showing the data simply doesn't suffice here. Furthermore, because the BG are important neural sources of song motor variability necessary to vocal motor learning, we need to know whether trial to trial variability is affected by DCN lesions in either juvenile or adult birds. Adult DCN lesion data must be shown and analysis of motor effects (including CV of pitch) in juveniles and adults should be included. Adult pitch learning experiments would help this study a lot although they are not required.

Related to this, in some of the juveniles, the cerebellar lesions seem to not just block further learning but produce *impairments* relative to pretraining. In other words, not only do they stop learning, but they forget what they already learned or are unable to express it. Therefore, it would be useful to see an analysis of the birds’ songs soon after the lesion, as well as many days later in the crystallized period.

1f) The authors acknowledge that effects of cerebellar lesions on non-song related motor impairments could have indirectly affected song learning, and that they will address this more specific lesions of cerebello-thalamic projections in future studies. In the meantime, to interpret the current results, it would be helpful to have more information about the behavior of the lesioned animals. Did they sing as many times per day between lesion and testing as the sham controls? This could affect the rate of learning. Also, it would be helpful to consider as controls any animals where the cerebellar lesions might have missed the lateral DCN target. Were there any animals with motor deficits, but which learned to sing ok?

2) The specificity of the targeting of pharmacological manipulations and recordings to specific brain areas must be addressed. This set of concerns might be largely addressed by providing a consensus map of where recordings/injections were made relative to the boundaries of Area X, and by conducting control experiments to measure the spread of the inactivation from DTZ to DLM.

2a) The paper to some extent replicates earlier findings from Person et al. However, the anatomical data presented here are much less conclusive. One of the bigger problems is that injections into area X clearly spread into the surrounding tissue (Figure 1A). Because area X is surrounded by other basal ganglia structures, the resulting tracing might be entirely due to general motor-related (non-song) pathways from the cerebellum to the basal ganglia. Given the imprecision of these injections, this leads to the question of whether the reported electrophysiological recordings were in area X, or whether some of the recordings were similarly in the non-song parts of the basal ganglia.

2b) In demonstrating transmission through DTZ, the authors acknowledge that the drug could've spilled into the nearby DLM. To control for this, they inactivate LMAN. However, unlike the thalamic nuclei, LMAN is very large, has variable stereotaxic coordinates across animals, and is non-trivial to inactivate in its entirety. No evidence is provided to show the completeness of these inactivations. To make matters worse, projections from LMAN to area X are topographic, so missing even a small part of LMAN could leave unaffected pockets of area X. It is reasonable to think that the authors are selectively hitting these pockets with their electrodes because they are using extracellular recordings that are biased toward particularly active cells.

2c) In several cases (e.g. Figure 5 and Figure 6) drug washout traces are profoundly different from control traces, raising the possibility that some of the observed effects are due to decreased health of the tissue or general condition of the animal under anesthesia.

2d) When the blockade of excitatory transmission at a given site reduces the effects of cerebellar stimulation at another site in the circuit, it could be because the signals from the cerebellum are transmitted through the nucleus where excitation was blocked. However, it also could be that the pharmacological manipulation has nonspecifically reduced the tonic drive to the site being recorded, making it less excitable, and hence less responsive to the cerebellar stimulation, even if that stimulation is not transmitted via the site of the pharmacological manipulation. One approach to address this might be to report effects of the pharmacological manipulation on the basal firing rate at the site of recording.

2e) The description of unit data collected from Area X could be improved, with caveats regarding cell type identification. The firing rates of their pallidal-like neurons are generally low, sometimes as low as 20 Hz, suggesting that they may include non pallidal-like neurons. It would be informative to show the raw spontaneous firing rate of the data so that one can know the rough estimate of the proportion of pallidal-like neurons in their data. It would be helpful if the authors could provide insight into the functional innervation of non-pallidal cells by the DCN, although I appreciate that this may be hard to pull out. Show more centrally targeted injections into Area X, as DTZ projection to surrounding striatum (outside of Area X) cannot be fully excluded by the example shown in Figure 1.

[Editors' note: further revisions were requested prior to acceptance, as described below.]

Thank you for resubmitting your work entitled "A subcortical circuit linking the cerebellum to the basal ganglia engaged in vocal learning" for further consideration at *eLife*. Your revised article has been reviewed by Andrew King (Senior Editor), a Reviewing Editor, and two reviewers.

This manuscript describes the functional anatomy supporting a cerebellar contribution to birdsong learning. Using electrical stimulation of the cerebellar deep nuclei, combined with single unit or multi-unit recording from different nuclei of the song system and pharmacological perturbations, the authors lay out a candidate pathway for the cerebellum to influence activity throughout the song learning circuitry. The manuscript has been extensively revised to address the comments raised in the previous review. However, there are some remaining issues that need to be addressed before acceptance, as outlined below in the reviewer comments. In particular, they would like you to address the issue of functional connectivity between LMAN and Area X.

Reviewer #2:

The authors' response to prior concerns is overall quite constructive and the manuscript is improved. I remain a bit underwhelmed by the effects of DCN lesions on song learning, but the additional analysis strengthens the conclusion that there is some (relatively weak) effect. Here I would point to Figure 7—figure supplement 2 where all the lesioned birds show, at least to my eye, more similarity to their respective tutors than they do to one another; and Figure 7—figure supplement 3B (mislabeled in the legend as F), where the ranges of the normalized similarity scores are almost completely overlapping. Again, I appreciate the extra work that the authors have done on this account, it just doesn't appear to be a very big effect (which is OK, perhaps the role of the DCN is more subtle than that of the basal ganglia, or DCN lesions can be somewhat compensated for by the rest of the song circuitry).

The adult behavioral experiments are a good addition. Although the effects do not reach significance, it does look like there is a similar trend in juveniles and adults where DCN lesions increase the mean and variance of syllable duration. I realize the adult effects do not achieve significance, but the trend is there nonetheless. I think that the authors are on track when they discuss the possibility that an acute insult to song timing could ultimately interfere with learning.

The physiology experiments are sound and the addition of the description of the gabazine experiments in Area X (relating to Figure 5G-I) substantially strengthen the conclusion that the DCN is functionally connected to Area X and can also influence downstream regions, including LMAN and RA. I would recommend that the authors state that the physiological recordings were made under (isoflurane) anesthesia when they begin to describe the results of these experiments (subsection “The connection from DCN to basal ganglia is functional”). I would also suggest that they qualify their findings in a similar manner when they discuss modulation of Area X by the DCN (Discussion section). It remains to be seen whether and how the DCN modulates activity in the cortico-basal ganglia network in singing birds.

I found this version of the manuscript somewhat harder to read than the original version, perhaps because of all the material that was added to satisfy reviewers' concerns. The physiology section is quite long and although well done represents experimental variations on a common theme. If DCN can modulate pallidal cells, then the LMAN and RA effects are not too surprising. I am not recommending that these results be excluded, because they do show a DCN influence on premotor neurons, something that is arguably harder to demonstrate in rodents and primates. But I wonder whether the LMAN and RA data could be collapsed into a single figure?

The writing could use some work. I point out a few places where editing will help, but it would be good to subject the manuscript to a couple of rounds of tightening.

Reviewer #3:

In this revised manuscript, Pidoux et al., have considerably expanded their investigation of how cerebellar circuits contribute to vocal learning in the zebra finch. Specifically, the authors have focused their attention on more careful quantification and analysis of behavior following acute cerebellar DCN lesions during the sensorimotor phase of song learning, now revealing a discrete effect of these lesions on syllable duration. These results are in line with the role of the cerebellum in regulating motor timing, and thus provide important insight into the specific contribution of cerebellar output to vocal learning that was absent in the initial submission. Based on these additional behavioral analyses, along with significant modifications to address previous technical concerns regarding electrode placement and pharmacological inactivation, the revised manuscript effectively accounts for my concerns that arose following the initial submission. By integrating another key component of the vocal learning circuitry into the network of well-studied cortical and basal ganglia circuits involved in song learning and identifying a role for cerebellar circuits in regulating how song timing is learned, this manuscript now represents a significant advance in the field of sensorimotor learning. I thus have only a few additional points:

1) If DCN stimulation activates LMAN with short latency (Figure 5), and LMAN projects to Area X, why doesn't LMAN inactivation alter Area X pallidal neuron spiking in any way (Figure 4)? Is this because the specific pallidal cells that project to DLM (that can activate LMAN when silenced) don't receive input back from LMAN? Even if LMAN inputs primarily go to the medium spiny cells, shouldn't one expect some effect on Area X pallidal cell spiking? If this prediction is correct, then the data are concerning. If not, however, for those who do not specialize in birdsong circuitry, it would be extremely helpful to more explicitly articulate the rationale here, as the naïve prediction is that LMAN activity should impact Area X spiking, and thus Area X spiking should change when LMAN is inactivated.

---

## [Author Response]

[Editors' note: the authors’ plan for revisions was approved and the authors made a formal revised submission.]

Essential revisions:1) All four reviewers agreed that the behavioral studies are where the greatest potential significance of the study lies and also where the experimental evidence, as presented, falls farther from the mark. Only 10 random songs were analyzed pre- and post-lesion and are summarized with a single similarity score. More extensive analysis of the behavioral results is crucial.

We agree with the reviewers that our initial analysis of song similarity was minimal and now extended it to strengthen the claim that the cerebellum significantly contributes to song learning. In detail, we have:

We have applied a recently published method for the analysis of song similarity between juvenile zebra finches and their tutor (Mandelblat-Cerf and Fee, 2014), in addition to our custom-based spectral cross-correlation analysis (now in Figure 7—figure supplement 3). We have also run the classical software developed by Tchernichovski et al. (SAT in Matlab) but given its lower sensitivity we chose to present only the results obtained with the method from Mandelblat-Cerf and Fee (2014). This analysis is now presented in Figure 7D-E-F, with a description of the associated methods in subsection “Song imitation analysis” and the results in subsection “DCN lesion impairs song learning in juvenile zebra finches”.

The similarity analysis was applied to a larger data pool consisting in 50 to 100 song bouts (each including several song motifs) from two consecutive days in each condition. We have carefully curated this data set to avoid artefactual contamination of the data by any nonsong audio signal (cage noises, calls or any other vocal activity).

Most importantly, we now present a complementary analysis of the acute effects of DCN lesions on song (comparing song features such as duration, pitch or amplitude and their variability before and after lesion and at crystallization). This analysis is presented in Figure 8, Figure 8—figure supplement 1 and Figure 8—figure supplement 2, with the methods described in subsection “Song temporal features, fundamental frequency and amplitude analysis”, and the results described in subsection “DCN lesions affects song temporal features in juvenile birds”.

An important point to keep in mind at this stage is that DCN lesions were applied at 60 days post hatch (dph), which is in the middle of the sensorimotor learning period. We initially made this deliberate choice to be able to compare motif syllables before and after lesions (syllables tend to be hard to recognize before 60dph as the babbling birds do not display any repeated motif structure). However, part of the learning process has already occurred at that stage and an increase in similarity between 60dph and crystallization (90 dph) is surprisingly difficult to reveal using classical song analysis methods. Our attempts to perform lesions on younger birds were disappointing as mortality was very high among subjects. The integrity of the lateral DCN may be crucial in young bird for survival skills like seed cracking or perching.

1a) The authors lesion the DCN in juvenile birds using either chemical or electrolytic methods. They report that learning (copying?) of a tutor song is impaired by these lesions. Several problems arise, though, in part because they employ a novel analysis method in place of rather than in addition to acoustic analysis methods that are standard in the field (i.e., SAP). It is unclear how they calculated similarity score (is it the peak of the cross-correlation, as suggested in subsection “DCN lesion impairs song learning in juvenile zebra finches”, or is it really the sum of the cross-correlation as shown in subsection “Song analysis”? In either case, if the score is based on the simple normalized cross correlation, it would be difficult to interpret their data because the score can be very low when juveniles improvise the gap/syllable durations or syllable orders, which often happens in normal zebra finches. A parallel analysis of song data using more established methods and showing more examples of spectrograms of the adult songs of birds with juvenile DCN lesions would be necessary for reviewers and other readers to assess the significance of the behavioral outcomes. The authors should also show the raw similarity score at 60 dph because there should be large variability in learning already at the age, which could significantly affect the normalized similarity score at 90 dph.

We have applied a recently published method for the analysis of song similarity between juvenile zebra finches and their tutor (Mandelblat-Cerf and Fee, 2014), in addition to our custom-based spectral cross-correlation analysis (now in Figure 7—Figure supplement 3). We have also run the classical software developed by Ofer Tchernichovski and colleagues (SAT in Matlab) but given its lower sensitivity we chose to present only the results obtained with the method from Mandelblat-Cerf and Fee, (2014). This analysis is now presented in Figure 7D-E-F, with a description of the associated methods in subsection “Song temporal features, fundamental frequency and amplitude analysis” and the results in subsection “DCN lesion impairs song learning in juvenile zebra finches”.

We have also added a supplementary figure with several example spectrograms of the adult songs of birds with juvenile DCN lesions (Figure 7—Figure supplement 2).

1b) Is there a specific contribution of this cerebellar circuit to vocal learning? In the final figure, the authors demonstrate that lesions of the cerebellar dentate nucleus impair the ability of juvenile birds to copy their tutor's song using a similarity score metric. It is unclear, however, whether there are any specific differences in the learned song resulting from cerebellar lesion that would not be seen by lesioning other brain areas involved in song learning. The manuscript would be greatly enhanced if the authors could perform further analyses designed to identify whether any discrete song features are selectively impaired by cerebellar lesions. For example, is there any specific temporal or pitch-related deficit in song production following tutoring? Is the trajectory of learning altered? Is there any evidence based on learning trajectory that learning is simply slowed, and thus incomplete at the time of measurement? Such information would be extremely valuable in order to compare these experiments with the learning deficits that result from lesions to other brain areas involved in vocal learning and could help guide meaningful hypotheses about the specific role of cerebellum in vocal learning.

We have now included the analysis of the acute effects of DCN lesions on discrete song features including fundamental frequency (also called pitch), duration and sound amplitude. We have compared the changes undergone by these song features before and after surgery, with multiple time points following surgery to account for transient and long-term effect of lesions. We also report the value of these features at crystallization, to determine the learning trajectory and how it is affected by the lesions. This analysis is presented in Figure 8, Figure 8—figure supplement 1 and Figure 8—figure supplement 2, with the methods described in subsection “Song temporal features, fundamental frequency and amplitude analysis”, and the results described in subsection “DCN lesions affects song temporal features in juvenile birds”.

The main result of this analysis is that syllable duration is acutely affected by DCN lesion. This effect may help understanding how the cerebellum contributes to song learning: with fine adjustment of syllable duration, in line with the well documented role of the cerebellum in motor timing. The learning trajectory for syllable duration seems to be affected in juveniles with a DCN lesion as well, as the change in duration between surgery and crystallization is greater in the sham group than in the lesion group (see subsection “DCN lesions affects song temporal features in juvenile birds”). As lesions were made relatively late in the learning process, the evolution of song between 55-60 dph (lesions) and 90 dph (crystallization) is relatively small and intermediate time points were therefore not included.

1c) The description of the timing of the DCN lesions is critical to interpreting what type of learning – sensory versus sensorimotor – the DCN may be contributing to. The timeline in Figure 7 suggests that DCN lesions may be made late enough in development to largely affect song copying rather than tutor song memorization. Language in the Results section and Discussion section should be precise when referring to song copying versus sensory learning. The authors should discuss whether their experimental design allows them to assign the effects of DCN lesions more exclusively to song copying. This is a place where more description of the experimental design in the results would help a general audience.

We agree with the reviewers that we did not emphasize sufficiently this aspect of the protocol in the previous version of our manuscript. Lesions were purposefully performed after the end of the sensory learning period, and our results do not infer any function of cerebellar circuit in the sensory learning process (memorization of tutor song). Rather, we are probing the role of the cerebellum in the sensorimotor (song copying) process. We have now highlighted this point in the Results section before describing the behavioral effects of the lesions, and in the Discussion section.

1d) The central conclusion of the paper – that DCN is involved in song learning – rests on comparing DCN-lesioned and sham-lesioned birds. The authors claim that these two groups are different because sham-lesioned birds imitate the tutor with a p value of 0.04, while DCN-lesioned birds "don't imitate" with a p value of 0.06. This kind of egregious misuse of statistics is the reason many results published in the literature cannot be replicated! Comparing p values of two groups is simply not valid; for instance, these values could be different because of differences in sample sizes or differences in sample variances. The only way to compare two conditions is by comparing the data directly in a two-sided statistical test. Sorting the raw values in Figure 7F makes it clear that the two populations are not actually statistically different. Given the high quality of imitation in some of the DCN-lesioned birds, it appears that the opposite conclusion may be correct – that DCN is not necessary for learning.

We agree with the reviewers that our interpretation of the statistical test was, in retrospect, misleading. We believe that conducting an additional similarity analysis (from Mandelblat-Cerf et al., 2014) has strengthened our point that the effect of DCN lesion on juvenile song imitation is significant. To reveal this effect by comparing the lesion group to the sham group, we had to separate the lesion group in two: birds with significant lesions (<75% lateral DCN left) and birds with very small lesions (>75% left). This allowed us to exclude 3 birds that had a very small lesion from the similarity analysis, and that led to a significant difference in the imitation score between the large lesion group and the sham group. In addition to the correlation between lesion size and imitation score (both in our custom-made analysis of similarity and in the Mandelblat-Cerf and Fee method), we believe that these results now strongly support a role of the cerebellum in juvenile song learning.

1e) It is inadequate to state in subsection “DCN lesion impairs song learning in juvenile zebra finches” that DCN lesions in adults had no effects on song and then not show any supporting data. Finding that the DCN only affects learning or also affects adult performance might be equally interesting. But not showing the data simply doesn't suffice here. Furthermore, because the BG are important neural sources of song motor variability necessary to vocal motor learning, we need to know whether trial to trial variability is affected by DCN lesions in either juvenile or adult birds. Adult DCN lesion data must be shown and analysis of motor effects (including CV of pitch) in juveniles and adults should be included. Adult pitch learning experiments would help this study a lot although they are not required.Related to this, in some of the juveniles, the cerebellar lesions seem to not just block further learning but produce *impairments* relative to pretraining. In other words, not only do they stop learning, but they forget what they already learned or are unable to express it. Therefore, it would be useful to see an analysis of the birds songs soon after the lesion, as well as many days later in the crystallized period.

We have now included the analysis of the acute effects of DCN lesions on discrete song features and their variability, including fundamental frequency (also called pitch), duration and sound amplitude. We have compared the changes undergone by these song features and their variability before and after surgery both in adults and juveniles. All the data is now shown either in the main figures (Figure 8) or in the supplements (Figure 8—figure supplement 1 and Figure 8—figure supplement 2). To highlight transient and long-term effects of the lesion, we included multiple time points following surgery and also report the value of these features and their CV at crystallization. This analysis is presented in Figure 8, Figure 8—figure supplement 1 and Figure 8—figure supplement 2, with the methods described in subsection “Song temporal features, fundamental frequency and amplitude analysis”, and the results described in subsection “DCN lesions affects song temporal features in juvenile birds”.

The main result of this analysis is that syllable duration is acutely affected by DCN lesion while fundamental frequency and amplitude, or their variability, are not affected. This effect may help understanding how the cerebellum contributes to song learning: with fine adjustment of syllable duration, in line with the well documented role of the cerebellum in motor timing. The learning trajectory for syllable duration seems to be affected in juveniles with a DCN lesion as well, as the change in duration between surgery and crystallization is greater in the sham group than in the lesion group (see subsection “DCN lesions affects song temporal features in juvenile birds”). As lesions were made relatively late in the learning process, the evolution of song between 60dph (lesions) and 90dph (crystallization) is relatively small and intermediate time points were therefore not included.

1f) The authors acknowledge that effects of cerebellar lesions on non-song related motor impairments could have indirectly affected song learning, and that they will address this more specific lesions of cerebello-thalamic projections in future studies. In the meantime, to interpret the current results, it would be helpful to have more information about the behavior of the lesioned animals. Did they sing as many times per day between lesion and testing as the sham controls? This could affect the rate of learning. Also, it would be helpful to consider as controls any animals where the cerebellar lesions might have missed the lateral DCN target. Were there any animals with motor deficits, but which learned to sing ok?

We agree that side-effects (non-song-specific) of the lesions could make the interpretation of the data more difficult. We have now included a quantification of the rate of singing in all animals following lesion (or sham surgery in sham birds), and we did not find any significant effect of the DCN lesion of the rate of singing following surgery. This result is represented in Figure 7—Figure supplement 1. Our extensive analysis of song imitation now reveals that the 3 birds with very small lesion sizes (>75% lateral DCN left) tend to have a better similarity score and had to be excluded from the lesion group to reveal a significant difference with the sham group. We believe that this segregation between large and small lesions, in combination with our analysis of the correlation between lesion size and similarity, is now clearly showing that birds with smaller DCN lesions have little effect on song imitation. We did not quantify general motor deficit (difficult as there is no clear clinical rating for these birds) in these birds however.

2) The specificity of the targeting of pharmacological manipulations and recordings to specific brain areas must be addressed. This set of concerns might be largely addressed by providing a consensus map of where recordings/injections were made relative to the boundaries of Area X, and by conducting control experiments to measure the spread of the inactivation from DTZ to DLM.

As suggested by reviewers, we have now added a consensus map of where recordings were performed in Area X (Figure 9A). Moreover, we did a control experiment quantifying the typical distance for the diffusion of NBQX/APV. To facilitate this control, we quantified the percentage of response in Area X following drug injection at different distance from the recorded neurons. Results are summarized in Figure 9C and we refer to this experiment in the subsection Pharmacology”.

2a) The paper to some extent replicates earlier findings from Person et al. However, the anatomical data presented here are much less conclusive. One of the bigger problems is that injections into area X clearly spread into the surrounding tissue (Figure 1A). Because area X is surrounded by other basal ganglia structures, the resulting tracing might be entirely due to general motor-related (non-song) pathways from the cerebellum to the basal ganglia. Given the imprecision of these injections, this leads to the question of whether the reported electrophysiological recordings were in area X, or whether some of the recordings were similarly in the non-song parts of the basal ganglia.

We apologize for the confusion due to the bad placement of the CTB injection in Area X. We now provide additional data confirming X-specific projection of DTZ neurons. Unfortunately, our recent CTB-labelling experiment were inconclusive (due to low fluorescence levels), but we replicated the experiment with classical dextran-conjugated dyes to confirm the DTZ-X projection of neurons in close proximity of cerebellar projections. Most importantly, a recent paper by the group of S Sober has very nicely and carefully replicated the finding by Person et al., with more sophisticated and precise techniques for anatomical tracing. Given that this is now published, we do not believe that additional anatomical tracing would add value to the present manuscript.

We would like to emphasize that this bad placement in one anatomical experiment does not question our general ability to locate Area X however. Importantly, we found a response to DCN stimulation in all pallidal cells recorded on an experiment where DCN stimulation could trigger at least one response, so it is virtually impossible that DCN stimulation could evoke a response outside of Area X but not in X. And histological reconstruction of the electrode penetration trajectories allowed confirmation of the location of recording sites in Area X.

2b) In demonstrating transmission through DTZ, the authors acknowledge that the drug could've spilled into the nearby DLM. To control for this, they inactivate LMAN. However, unlike the thalamic nuclei, LMAN is very large, has variable stereotaxic coordinates across animals, and is non-trivial to inactivate in its entirety. No evidence is provided to show the completeness of these inactivations. To make matters worse, projections from LMAN to area X are topographic, so missing even a small part of LMAN could leave unaffected pockets of area X. It is reasonable to think that the authors are selectively hitting these pockets with their electrodes because they are using extracellular recordings that are biased toward particularly active cells.

We agree with the reviewers that pharmacological effects are unlikely to spread into the whole volume of LMAN. In fact, we now show that the drug spread is around 200 μm. While this diffusion distance is smaller than LMAN radius (250-350 μm), it is not very far from it and we therefore believe that drug injections have exerted an influence on a substantial part of the nucleus.

Moreover, several indications can rule out such a drastic “missing” of LMAN as an important relay of responses in Area X. First of all, our extracellular recordings are not biased toward responsive cells (responding to DCN stimulation), but toward spontaneously active cells, in particular the ones with spontaneous activity >25Hz (hypothesized to be pallidal cells). Blocking excitation within LMAN (the manipulation we apply) does not change much the local spontaneous activity (result provided in Figure 10). Moreover, the spontaneous activity of LMAN neurons is not the main drive of pallidal spontaneous activity (these are spontaneously active in a slice). Thus, it is very unlikely that blocking excitation in LMAN significantly affects the spontaneous activity of Area X neurons. Regarding the topography of the projection, we agree that pharmacological manipulation in a small volume in LMAN will only perturb transmission to a small zone in Area X. However, we have shown that our injections cover a significant part of LMAN. Finally, our ability to block LMAN responses by pharmacologically blocking excitation in Area X (bigger than LMAN) with the same technique suggest that the reverse should be just as plausible.

2c) In several cases (e.g. Figure 5 and Figure 6) drug washout traces are profoundly different from control traces, raising the possibility that some of the observed effects are due to decreased health of the tissue or general condition of the animal under anesthesia.

We agree with the reviewers that washout traces often differ from control traces. While the main characteristics of the responses were recovered in the washout (polarity, overall strength, latency of the first response), it is very common that the washout process is progressive and displays different dynamics for the various phases of the response. We believe that such slow and inhomogeneous recovery should be expected from a high concentration drug solution expelled from a single site and slowly diffusing into the target nucleus before being slowly washed out. There is no reason to believe that the drug washout is homogenous over the extent of the target nucleus, and inhomogeneity in the spatial extent of drug concentration itself may explain the different shapes of washout responses. We do not believe that differences in the specific response profiles between washout and control impair the interpretation of our results, if the main characteristics of the response are conserved (again: polarity, overall strength, latency of the first response).

Importantly, responses to DCN stimulation could be sustained for several hours in many neurons, and effects of the general condition of the animal under anesthesia typically impact spontaneous activity as strongly as the response to stimulation. We now provide a detailed analysis of spontaneous activity during control, drug condition and washout for each pharmacological manipulation (Figure 10) and refer to this experiment in the subsection “Pharmacology”.

2d) When the blockade of excitatory transmission at a given site reduces the effects of cerebellar stimulation at another site in the circuit, it could be because the signals from the cerebellum are transmitted through the nucleus where excitation was blocked. However, it also could be that the pharmacological manipulation has nonspecifically reduced the tonic drive to the site being recorded, making it less excitable, and hence less responsive to the cerebellar stimulation, even if that stimulation is not transmitted via the site of the pharmacological manipulation. One approach to address this might be to report effects of the pharmacological manipulation on the basal firing rate at the site of recording.

We agree with the reviewers that a reduced spontaneous drive to a neuronal population could affect their ability to respond to a different set of inputs. To resolve this issue, we now provide a detailed analysis of spontaneous activity during control, drug condition and washout for each pharmacological manipulation (Figure 10) and we refer to this experiment in the subsection “Pharmacology”.

2e) The description of unit data collected from Area X could be improved, with caveats regarding cell type identification. The firing rates of their pallidal-like neurons are generally low, sometimes as low as 20 Hz, suggesting that they may include non pallidal-like neurons. It would be informative to show the raw spontaneous firing rate of the data so that one can know the rough estimate of the proportion of pallidal-like neurons in their data. It would be helpful if the authors could provide insight into the functional innervation of non-pallidal cells by the DCN, although I appreciate that this may be hard to pull out. Show more centrally targeted injections into Area X, as DTZ projection to surrounding striatum (outside of Area X) cannot be fully excluded by the example shown in Figure 1.

We now provide the distribution of spontaneous firing rate of recorded neurons in Area X (Figure 9B). Unfortunately, we were not able to perform the initially suggested supplementary experiments to juxtacellular label of recorded cells in Area X. However, given the unimodal distribution of spontaneous firing rates, we believe that we have recorded from a homogeneous population corresponding to pallidal neurons. the strongly bimodal distribution of spontaneous activity observed in our recording (see Materials and methods section) and the relative scarcity of neurons displaying a low spontaneous activity in the song-related basal ganglia nucleus (Farries and Perkel, 2002), our dataset is likely to contain mostly if not only pallidal neurons. A contribution from a small fraction of spontaneous striatal interneurons cannot, however, be ruled out in the absence of post-hoc histological verification of the recorded cell type. We have added a sentence in the discussion to highlight the fact that we cannot confirm the nature of the recorded neurons (subsection “Similarities and differences between the cerebello-thalamo-basal ganglia pathways of mammals and songbirds”).

[Editors' note: further revisions were requested prior to acceptance, as described below.]

This manuscript describes the functional anatomy supporting a cerebellar contribution to birdsong learning. Using electrical stimulation of the cerebellar deep nuclei, combined with single unit or multi-unit recording from different nuclei of the song system and pharmacological perturbations, the authors lay out a candidate pathway for the cerebellum to influence activity throughout the song learning circuitry. The manuscript has been extensively revised to address the comments raised in the previous review. However, there are some remaining issues that need to be addressed before acceptance, as outlined below in the reviewer comments. In particular, they would like you to address the issue of functional connectivity between LMAN and Area X.Reviewer #2:The authors' response to prior concerns is overall quite constructive and the manuscript is improved. I remain a bit underwhelmed by the effects of DCN lesions on song learning, but the additional analysis strengthens the conclusion that there is some (relatively weak) effect. Here I would point to Figure 7—figure supplement 2 where all the lesioned birds show, at least to my eye, more similarity to their respective tutors than they do to one another; and Figure 7—figure supplement 3B (mislabeled in the legend as F), where the ranges of the normalized similarity scores are almost completely overlapping. Again, I appreciate the extra work that the authors have done on this account, it just doesn't appear to be a very big effect (which is OK, perhaps the role of the DCN is more subtle than that of the basal ganglia, or DCN lesions can be somewhat compensated for by the rest of the song circuitry).

We agree with reviewer #2 that the effect of our DCN lesion on song learning, although significant, are relatively weak, in particular compared to effects induced by lesions in other parts of the song system. We would like however to highlight that given the non-specific and convergent circuits running through the DCN, we had to limit our lesions and most animals received lesion that left at least half of the lateral DCN intact, leaving lots of room for compensation. We tempered our conclusions with respect to the cerebellar contribution to song learning and highlighted possible compensatory mechanisms in the Results section and Discussion section. In particular, we changed the partial conclusion in subsection “DCN lesion impairs song learning in juvenile zebra finches”, which now clearly acknowledges the subtle behavioral effects:

“In conclusion, partial lesions in the lateral DCN induced a subtle but significant effect on the song acquisition process in juvenile zebra finches, providing evidence that the cerebellum contributes to song learning.”

We also shortened and simplified the discussion concerning small lesion size and possible compensation in subsection “Involvement of the cerebellum in timing processing”:

“Given the relatively small extent of the lesions performed and that other circuits in the song system may compensate for the effect of DCN lesions, we cannot exclude a cerebellar contribution to fundamental frequency.”

We changed the name of the Figure 7—figure supplement 3F to Figure 7—figure supplement 3B.

The adult behavioral experiments are a good addition. Although the effects do not reach significance, it does look like there is a similar trend in juveniles and adults where DCN lesions increase the mean and variance of syllable duration. I realize the adult effects do not achieve significance, but the trend is there nonetheless. I think that the authors are on track when they discuss the possibility that an acute insult to song timing could ultimately interfere with learning.

We would like to thank the reviewer to highlight this similarity and to suggest making the parallel, as we indeed believe that the same mechanism may be at play, although not reaching significance in adults. We now explicitly refer to the similar trend in subsection “DCN lesions affect song temporal features in juvenile birds”:

“In adult birds, the effect of DCN lesions on syllable duration did not reach significance, although a similar trend to increase the relative change in syllable duration compared to sham was observed (Figure 7—figure supplement 2A-B, Wilcoxon test, non-significant, see Table2 for detailed statistical value).”

The physiology experiments are sound and the addition of the description of the gabazine experiments in Area X (relating to Figure 5G-I) substantially strengthen the conclusion that the DCN is functionally connected to Area X and can also influence downstream regions, including LMAN and RA. I would recommend that the authors state that the physiological recordings were made under (isoflurane) anesthesia when they begin to describe the results of these experiments (subsection “The connection from DCN to basal ganglia is functional”). I would also suggest that they qualify their findings in a similar manner when they discuss modulation of Area X by the DCN (Discussion section). It remains to be seen whether and how the DCN modulates activity in the cortico-basal ganglia network in singing birds.

We now explicitly refer to the anaesthetized state of the animal in the paragraph concerning responses to stimulation in subsection “The connection from DCN to basal ganglia is functional”:

“We then determined whether this DCN-DTZ-Area X pathway drives activity within the basal ganglia. To this end, we investigated the responses evoked by DCN electrical stimulation in Area X neurons in anaesthetized zebra finches.”

Moreover, in the following sentence we clarify the type of anesthesia: “Most neurons are silent or display very little spontaneous activity in Area X under isoflurane anesthesia” (subsection “The connection from DCN to basal ganglia is functional”).

Finally, in the Discussion section, we also remind readers that we were working under anesthesia “Our data establish a functional excitatory projection from the lateral part of the DCN to the song-related basal ganglia nucleus Area X via a thalamic relay in DTZ in anaesthetized zebra finches.”

I found this version of the manuscript somewhat harder to read than the original version, perhaps because of all the material that was added to satisfy reviewers' concerns. The physiology section is quite long and although well done represents experimental variations on a common theme. If DCN can modulate pallidal cells, then the LMAN and RA effects are not too surprising. I am not recommending that these results be excluded, because they do show a DCN influence on premotor neurons, something that is arguably harder to demonstrate in rodents and primates. But I wonder whether the LMAN and RA data could be collapsed into a single figure?

We have revised the whole manuscript and tried our best to clarify and streamline our presentation of the results and the discussion, which had been extensively revised, sometime making the reading more difficult, as pointed out here.

We agree with reviewer #2 that the physiology section is quite long and contains repetitive results. We have now grouped results concerning responses in LMAN and RA, to DCN stimulation together in one single figure (now represented in Figure 5). As consequences, all figures after Figure 5 were also renamed references corrected all along the main text.

The writing could use some work. I point out a few places where editing will help, but it would be good to subject the manuscript to a couple of rounds of tightening.

We thank reviewer #2 for this careful editing of our manuscript and have made the suggested changes, as detailed here below. And awe revised the manuscript to the best of our ability to clarify the text, as indicated above.

Reviewer #3:In this revised manuscript, Pidoux et al., have considerably expanded their investigation of how cerebellar circuits contribute to vocal learning in the zebra finch. Specifically, the authors have focused their attention on more careful quantification and analysis of behavior following acute cerebellar DCN lesions during the sensorimotor phase of song learning, now revealing a discrete effect of these lesions on syllable duration. These results are in line with the role of the cerebellum in regulating motor timing, and thus provide important insight into the specific contribution of cerebellar output to vocal learning that was absent in the initial submission. Based on these additional behavioral analyses, along with significant modifications to address previous technical concerns regarding electrode placement and pharmacological inactivation, the revised manuscript effectively accounts for my concerns that arose following the initial submission. By integrating another key component of the vocal learning circuitry into the network of well-studied cortical and basal ganglia circuits involved in song learning and identifying a role for cerebellar circuits in regulating how song timing is learned, this manuscript now represents a significant advance in the field of sensorimotor learning. I thus have only a few additional points:1) If DCN stimulation activates LMAN with short latency (Figure 5), and LMAN projects to Area X, why doesn't LMAN inactivation alter Area X pallidal neuron spiking in any way (Figure 4)? Is this because the specific pallidal cells that project to DLM (that can activate LMAN when silenced) don't receive input back from LMAN? Even if LMAN inputs primarily go to the medium spiny cells, shouldn't one expect some effect on Area X pallidal cell spiking? If this prediction is correct, then the data are concerning. If not, however, for those who do not specialize in birdsong circuitry, it would be extremely helpful to more explicitly articulate the rationale here, as the naïve prediction is that LMAN activity should impact Area X spiking, and thus Area X spiking should change when LMAN is inactivated.

We agree with reviewer #3 that it can be somewhat surprising not to see a significant effect of LMAN inactivation on Area X responses to DCN stimulation as we have shown that stimulation effects propagate through the BG-cortical loop and evoked excitation in LMAN. Several reasons could explain this lack of effect. Firstly, fast responses are a minority in LMAN and most responses are quite late (latencies >100ms) so it is likely that most excitatory feedback to X due to propagation along the loop only arrives in Area X after the end of its first excitatory response. It is therefore unlikely to contribute to the peak response observed in Area X as it is not synchronized with the main (first-hand) response). As indicated in subsection “Data analysis”, the response strength used for quantification over the population of responding neurons only reflects the strength of the first excitation peak. Therefore, reverberation through the loop, which may induce a much slower excitation in pallidal neurons, does not contribute. Secondly, LMAN will induce a mix of excitation (direct excitation of pallidal neurons by LMAN projection neurons) and inhibition (through feedforward inhibition in Area X), and the sum of these effects may add-up to a very small global effect if it mixes up due to the various timing involved. Finally, we see a reduction in all but one pallidal neurons, which is very small and not significant, but may indicate that some small reverberated activity through LMAN contributes (only slightly) to the measured response strength. We modified the results to explain the lack of effect on Area X response strength and peak.

“While LMAN does not appear to mediate the main response to DCN stimulation in Area X pallidal neurons, it may participate to a reverberation of the responses through the Area X – DLM – LMAN loop. In this respect, is interesting to note that all but one pallidal neurons underwent a slight decrease in their response upon glutamatergic blockade in LMAN, possibly reflecting a reduced reverberation in the loop. As the measured response strength only reflects the first peak of excitatory response in Area X, the slow response mediated by the propagation through the loop is unlikely to provide an important contribution to this measure (see Methods).” (subsection “LMAN does not mediate Area X responses to DCN stimulation”)